# Classification and genetic targeting of cell types in the primary taste and premotor center of the adult *Drosophila* brain

**Gabriella R Sterne**[1,2]*, **Hideo Otsuna**[2], **Barry J Dickson**[2,3], **Kristin Scott**[1]*

[1]University of California Berkeley, Berkeley, United States; [2]Janelia Research Campus, Howard Hughes Medical Institute, Ashburn, United States; [3]Queensland Brain Institute, University of Queensland, Queensland, Australia

**Abstract** Neural circuits carry out complex computations that allow animals to evaluate food, select mates, move toward attractive stimuli, and move away from threats. In insects, the subesophageal zone (SEZ) is a brain region that receives gustatory, pheromonal, and mechanosensory inputs and contributes to the control of diverse behaviors, including feeding, grooming, and locomotion. Despite its importance in sensorimotor transformations, the study of SEZ circuits has been hindered by limited knowledge of the underlying diversity of SEZ neurons. Here, we generate a collection of split-GAL4 lines that provides precise genetic targeting of 138 different SEZ cell types in adult *Drosophila melanogaster*, comprising approximately one third of all SEZ neurons. We characterize the single-cell anatomy of these neurons and find that they cluster by morphology into six supergroups that organize the SEZ into discrete anatomical domains. We find that the majority of local SEZ interneurons are not classically polarized, suggesting rich local processing, whereas SEZ projection neurons tend to be classically polarized, conveying information to a limited number of higher brain regions. This study provides insight into the anatomical organization of the SEZ and generates resources that will facilitate further study of SEZ neurons and their contributions to sensory processing and behavior.

*For correspondence:
sternegr@berkeley.edu (GRS);
kscott@berkeley.edu (KS)

**Competing interest:** The authors declare that no competing interests exist.

## Introduction

Elucidating the neural architecture that underlies sensorimotor transformations for behavior requires the ability to resolve and manipulate neural circuits with single-cell precision. With a tractable number of neurons and well-developed genetic tools, *Drosophila melanogaster* is an excellent animal in which to investigate the basic principles of sensorimotor processing. Recent electron microscopy datasets provide unprecedented synaptic resolution of approximately 100,000 neurons that comprise the adult *D. melanogaster* brain (*Scheffer et al., 2020*; *Zheng et al., 2018*). If coupled with resources that provide genetic access to single neurons, this detailed anatomy may be probed to study complex circuits underlying sensory processing and behavior.

The subesophageal zone (SEZ) of the adult insect brain plays a critical role in many sensory-driven behaviors. Defined as the brain tissue below the esophageal foramen, it is situated in a central location between the motor circuits of the ventral nerve cord (VNC) and the higher order brain regions of the supraesophageal zone (*Ito et al., 2014*). In *Drosophila* and other insects, the SEZ participates in many context-dependent motor actions, including feeding, grooming, and locomotion, with evidence suggesting that it is involved in action selection (*Bidaye et al., 2020*; *Flood et al., 2013*; *Gordon and Scott, 2009*; *Hampel et al., 2017*; *Hampel et al., 2015*; *Mann et al., 2013*; *Manzo et al., 2012*; *Marella et al., 2006*; *Tastekin et al., 2015*; *Wang et al., 2004*). It receives direct sensory input from axonal arbors of gustatory and mechanosensory peripheral neurons and indirect input

from pheromone-sensing neurons (*Ling et al., 2014*; *Thistle et al., 2012*; *Thorne et al., 2004*; *Toda et al., 2012*; *Wang et al., 2004*; *Weiss et al., 2011*; *Zhang et al., 2013*). Two major outputs of the SEZ are descending neurons that convey information to the VNC and motor neurons that control the movement of the proboscis and antennae (*McKellar et al., 2020*; *Namiki et al., 2018*; *Stocker et al., 1990*). Recent work has delineated fascicle and neuropil-based columnar domains in the SEZ that are identifiable throughout development and has mapped sensory substructures in the SEZ in larvae and adults (*Hartenstein et al., 2018*; *Kendroud et al., 2018*; *Miroschnikow et al., 2018*). Despite these advances, the exploration of the function of SEZ neurons has been hindered by the lack of genetic access to individual cell types.

Previous studies of the function of SEZ cell types have relied on broad GAL4 lines or stochastic methods, which do not provide reliable access to individual neurons. Recent efforts using the split-GAL4 method in *Drosophila* have provided genetic access to libraries of single neurons in other brain regions, including the mushroom body, central complex, and lateral horn (*Aso et al., 2014a*; *Dolan et al., 2019*; *Wolff and Rubin, 2018*). In this intersectional method, two different enhancers are used to independently drive expression of either the GAL4 transcriptional activation domain (AD) or DNA-binding domain (DBD). These domains heterodimerize through leucine zipper fragments and drive transgene expression restricted to the intersection of the two expression patterns (*Luan et al., 2006*). Thus, split-GAL4 reagents may be rationally designed if they are constructed using enhancers with known expression patterns. To systematically probe the cellular anatomy of the SEZ and to enable genetic dissection of SEZ neural circuits, we set out to create a library of genetic reagents to label individual SEZ cell types using the split-GAL4 method.

Here, we report the creation of 277 split-GAL4 lines that we collectively term the SEZ Split-GAL4 Collection. We estimate that this collection targets nearly one third of all neurons with cell bodies in the SEZ of the adult *Drosophila* brain. Morphological clustering of the identified cell types reveals six layered (anterior to posterior) and stacked (inferior to superior) domains of organization in the SEZ. Furthermore, polarity analysis shows that many SEZ interneurons have inputs and outputs on the same processes, whereas SEZ projection neurons tend to be classically polarized, with inputs and outputs located in clearly distinct regions of the neuronal arbors. Taken together, the genetic reagents described here provide a valuable resource to investigate how diverse sensory inputs are processed by local SEZ circuitry to control specific behaviors.

## Results
### The SEZ contains about 1700 neurons

We set out to determine the number of neuronal cell bodies in the adult SEZ to inform the generation and assessment of split-GAL4 lines. The SEZ contains four neuropil subregions: the gnathal ganglia (GNG), saddle (SAD), prow (PRW), and antennal mechanosensory and motor center (AMMC; *Figure 1A and B*; *Ito et al., 2014*). These SEZ subregions are composed of cells from the tritocerebral, mandibular, maxillary, and labial neuromeres, which are genetically defined by the expression of known homeobox-containing, neuromere-specific genes. In order to estimate the number of neurons in the SEZ, we assessed the number of neuronal cell bodies labeled by these neuromere-specific markers. We used a single-cell transcriptome atlas of the *D. melanogaster* brain (*Davie et al., 2018*) to determine the relative proportions of neurons expressing SEZ neuromere-specific markers. We also directly counted cell bodies labeled by available SEZ neuromere-specific drivers (*Simpson, 2016*) in individual *D. melanogaster* brains. We estimated total SEZ neuron number by converting proportions derived from the single-cell transcriptome atlas into neuron number estimates based on the direct counts.

We first examined the relative number of cells and neurons in each of the subesophageal neuromeres. We filtered single-cell RNA-sequencing data from the transcriptome atlas (*Davie et al., 2018*) to include only cells with detectable levels of any of three homeobox-containing transcription factors that are specifically expressed in the SEZ neuromeres: *Deformed* (*Dfd*) (mandibular and maxillary neuromeres), *Sex combs reduced* (*Scr*) (labial neuromere), or *labial* (*lab*) (tritocerebral neuromere) (*Hirth et al., 1998*; *Kumar et al., 2015*). Of the 56,902 high-quality cells represented in the atlas, 390 are *Dfd*-positive, 691 are *Scr*-positive, 134 are *lab*-positive, and 15 express both *Dfd* and *lab* (*Figure 1C*). Together, 1230 cells in the atlas express these neuromere-specific markers, with

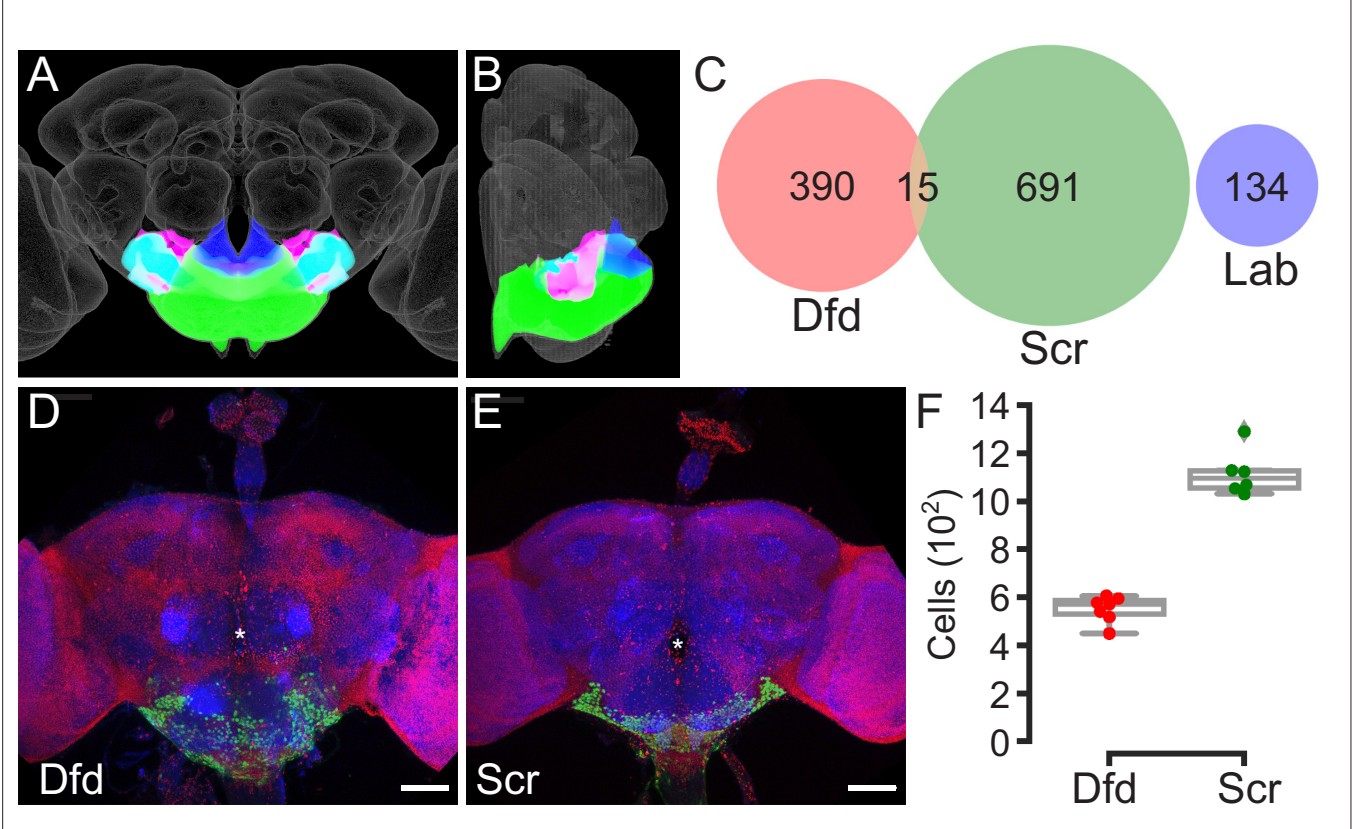

**Figure 1.** Estimating the number of cells in the subesophageal zone (SEZ). (**A, B**) Anterior (**A**) and medial (**B**) views of the central brain of the *Drosophila melanogaster* adult showing the location of the gnathal ganglia (GNG, green), saddle (SAD, fuchsia), antennal mechanosensory and motor center (AMMC, cyan), and prow (royal blue) in relation to the JRC 2018 unisex brain template (grey). Together, the GNG, SAD, AMMC, and PRW compose the SEZ. (**C**) Venn diagram of single cells with detectable *Dfd* (red), *Scr* (green), and/or *lab* (blue) as assessed with a single-cell transcriptome atlas. (**D, E**) Example overview images of the samples used to count the number of cells expressing Dfd-LexA or Scr-LexA. LexAop-nls-GCAMP6s (green) driven by Dfd-LexA (**D**) or Scr-LexA (**E**) in the adult central brain. All nuclei are labeled with His2Av-mRFP (red) and neuropil is labeled with nc82 (blue). Asterisks denote the location of the esophageal foramen. Scale bars, 50 μm. (**F**) Box plots displaying counts of cell bodies labeled by both His2Av-RFP and LexAop-nls-GCaMP6s when driven with Dfd-LexA (n = 7) or Scr-LexA (n = 6). Whiskers denote spread of samples within 1.5 interquartile range from the mean.

The online version of this article includes the following source data and figure supplement(s) for figure 1:

**Source data 1.** Dfd-LexA and Scr-LexA cell counts for panel F.

**Figure supplement 1.** Expression of labial-GAL4 in the adult central brain.

1182/1230 (96.1%) assigned to neuronal clusters based upon expression of neuronal genes. Although these results provide insight into relative numbers of cells in each subesophageal neuromere, they do not estimate neuron number in an individual brain because the single-cell RNA-sequencing atlas was constructed from multiple dissociated *D. melanogaster* brains.

To translate the proportions derived from single-cell RNA-sequencing data into an estimate of SEZ cell number, we directly counted cell nuclei in SEZ neuromeres in individual brains. Two knock-in LexA lines, Dfd-LexA and Scr-LexA (*Simpson, 2016*), were used to label cells in three of the four SEZ neuromeres: the mandibular and maxillary neuromeres and the labial neuromere, respectively (*Figure 1D and E*). We found that lab-GAL4, which is not a knock-in line, did not selectively label the tritocerebral neuromere, precluding cell counts of the fourth SEZ neuromere (*Figure 1—figure supplement 1*). Adult female brains expressing nuclear localized GCaMP6s driven by either Dfd-LexA or Scr-LexA (neuromere-specific) and histone tagged with red fluorescent protein (RFP) under the control of the tubulin promoter (all cells) were used for visualization and machine-learning-assisted quantification. Dfd-LexA labeled an average of 551 ± 54 cells (n = 7) while Scr-LexA labeled an average of 1115 ± 94 cells (n = 6) in the central brain (*Figure 1F*), generally consistent with the proportions seen in the

transcriptome atlas. Using the direct counts of Dfd-LexA cells to estimate total SEZ cell number based on the proportions derived from single-cell RNA-sequencing, we would expect ~1500–1850 cells in subesophageal neuromeres, ~1450–1750 (96.1%) of which are likely to be neurons. Using our Scr-LexA counts to estimate total SEZ number, we would expect ~1800–2100 cells, ~1700–2000 of which are likely to be neurons. These estimates are roughly consistent with previous estimates of secondary SEZ neuron number based on neuroblasts, which predicted ~2000 SEZ neurons (**Kuert et al., 2014**). We averaged the estimates based on Dfd and Scr counts to establish a final SEZ cell number estimate of ~1800 cells, of which ~ 1700 are neurons.

## The SEZ Split-GAL4 Collection provides genetic access to one third of all SEZ neurons

To characterize the morphology of individual SEZ cell types and to create a library of genetic reagents to provide specific access to these same cell types, we employed the split-GAL4 strategy (**Luan et al., 2006**). Since there is no consensus about how to define neuronal cell types, we relied on the stereo-typed morphology of *Drosophila* neurons to identify similar neurons across multiple samples. Cell types were defined as a pair or group of neurons with minimally variant morphology such that they were readily identifiable across multiple samples and driver lines by an expert. We used several strategies to identify novel SEZ cell types: (1) visual search through publicly available GAL4 collections (**Jenett et al., 2012**; **Pfeiffer et al., 2008**; **Tirian and Dickson, 2017**); (2) LexA-based MultiColor FlpOut (MCFO) single-cell labeling of Scr-LexA and Dfd-LexA; (3) MCFO screening of subsets of the Rubin and Vienna Tile (VT) GAL4 collections with dense SEZ expression (**Meissner et al., 2020**; **Nern et al., 2015**); and (4) re-registration of images of individual SEZ cell types from mosaic analysis of broad GAL4 drivers, available on FlyCircuit (**Chiang et al., 2011**). Ascending neurons (cell types with cell bodies in the VNC and outputs in the SEZ) were not included. In addition, AMMC neurons were not included as cell types in the AMMC have been analyzed extensively (**Matsuo et al., 2016**). Each novel cell type was given a unique (but not necessarily formulaic) name. Following cell type identification, we used the color depth maximum intensity projection (CDM) mask search tool (**Otsuna et al., 2018**) to select available split-halves that potentially labeled each cell type. After gathering a list of available split-halves likely to label a given cell type, we crossed all possible combinations of candidate ADs and DBDs and screened for split-GAL4 lines that specifically labeled the cell type of interest. The process of generating split-GAL4 lines acted as a built-in test of whether each cell type was properly defined. If a given cell type could not be reliably identified based on its stereotyped morphology, the chosen hemidrivers would be unlikely to intersect and the resulting lines would fail to label the targeted cell type. Only split-GAL4 lines that labeled targeted cell types are included in this collection.

We screened ~3400 split-GAL4 combinations using this strategy, which yielded 277 lines that provide precise access to single-cell types in the SEZ. The expression of each line is annotated to indicate the cell type that it was designed to target and the quality of the line (**Supplementary file 1**). These split-GAL4 lines label 138 SEZ cell types, 129 of which have not been previously reported. The quality of each line was rated as ideal (labeling only a single SEZ cell class and no other neurons in the brain or VNC), excellent (labeling the cell type of interest and 1–2 other cell types), good (labeling the cell type of interest and 3–5 other cell types), or poor (labeling the cell type of interest plus more than five other cell types). Amongst the 277 split-GAL4 lines that were generated, 63 are ideal, 86 are excellent, 99 are good, and 29 are poor (**Figure 2A**). An example line for each quality class is shown in **Figure 2B–E**. Poor lines are included in the collection if they improve genetic access to the target cell type as compared to existing GAL4 lines. Each target cell type is covered by at least one split-GAL4 line. The number and quality of split-GAL4 lines per targeted cell type is shown in **Figure 2F**.

To evaluate the completeness of coverage achieved by the SEZ Split-GAL4 Collection, we compared the total number of neuronal cell bodies covered by the split-GAL4 lines with our SEZ neuron number estimates. SEZ cell types fall into either unique or population classifications, where unique neurons encompass a single pair of cell bodies while population neurons are small groups of cell bodies with nearly identical arbors (**Namiki et al., 2018**). Therefore, one cell type may contribute one or multiple cell bodies per hemisphere. Taking this into consideration, the collection labels 510 neurons out of 1700 estimated, arguing that the SEZ Split-GAL4 Collection provides approximately 30% coverage of all SEZ neurons. In addition, 17 split-GAL4 lines specifically target SEZ motor neurons of the proboscis, totaling 36 cell bodies (**McKellar et al., 2020**). Moreover, the descending interneuron (DN) Split-GAL4

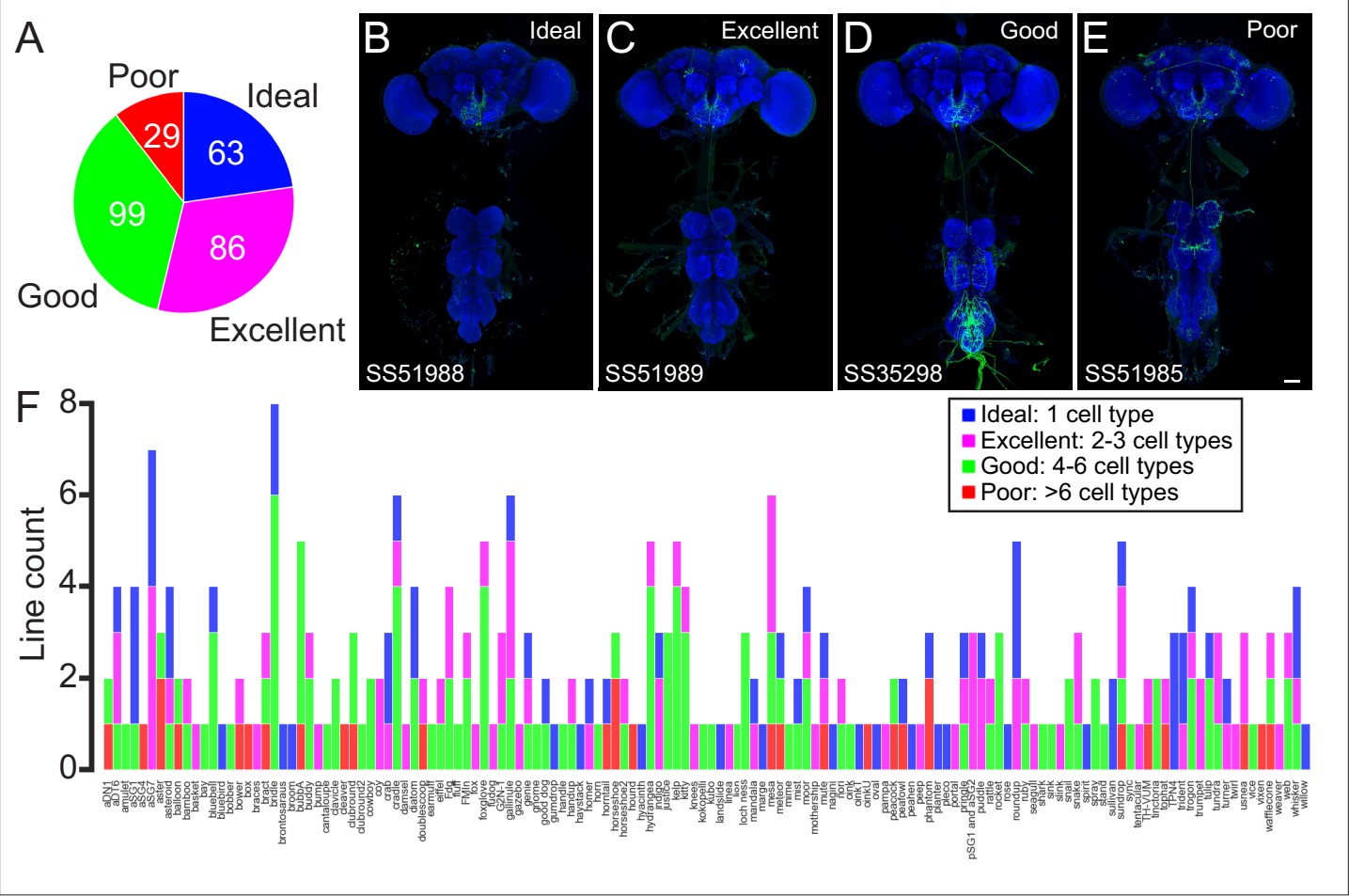

**Figure 2.** Quality of the lines in the subesophageal zone (SEZ) Split-GAL4 Collection. (**A**) The proportion of (royal blue), excellent (fuchsia), good (green), and poor (red) split-GAL4 lines included in the collection. (**B–E**) Examples of lines from each quality class. An example central nervous system is shown for each line. Expression pattern of the UAS reporter is shown in green, while neuropil is labeled with nc82 in blue. Scale bar is 50 μm. Each split-GAL4 line labels the same cell type, sundrop, but is of ideal (**B**), excellent (**C**), good (**D**), or poor (**E**) quality. (**F**) The number of ideal (royal blue), excellent (fuchsia), good (green), and poor (red) split-GAL4 lines included in the collection arranged by targeted neuron type.

Collection contains 41 DN cell types that comprise 242 additional cell bodies in the SEZ (out of 360 total DN cell bodies in the SEZ; *Namiki et al., 2018*). Together, the SEZ Split-GAL4 Collection, the proboscis motor neuron split-GAL4s, and the DN Collection provide precise access to 46% of SEZ neurons (788/1700). In summary, the SEZ Split-GAL4 Collection greatly improves genetic access to SEZ cell types, especially non-DN SEZ cell types. These split-GAL4 lines represent a substantial expansion of the knowledge of SEZ cell types and enable precise manipulation of the targeted cell types for behavioral, functional imaging, and morphological analyses. Confocal images of each line and instructions for requesting lines from the SEZ Split-GAL4 Collection can be found at https://splitgal4.janelia.org/.

## Clustering of SEZ cell types reveals six cellular domains

To investigate SEZ organization at a cellular level, we used the NBLAST algorithm to perform automated clustering of SEZ cell types to define cell type supergroups (*Costa et al., 2016*). NBLAST computes a pairwise neuronal similarity score by considering the position and local geometry of a query and target neuron. By comparing SEZ neurons with NBLAST in an all-by-all matrix, we clustered them into morphologically similar groups to reveal SEZ substructure. To prepare neuron imagery for the NBLAST algorithm, a single, unilateral example of each cell type was imaged at high resolution using MCFO and registered to a common unbiased template (*Bogovic et al., 2020*; *Nern et al., 2015*). Each cell type example was then segmented, skeletonized, and presented on the right side

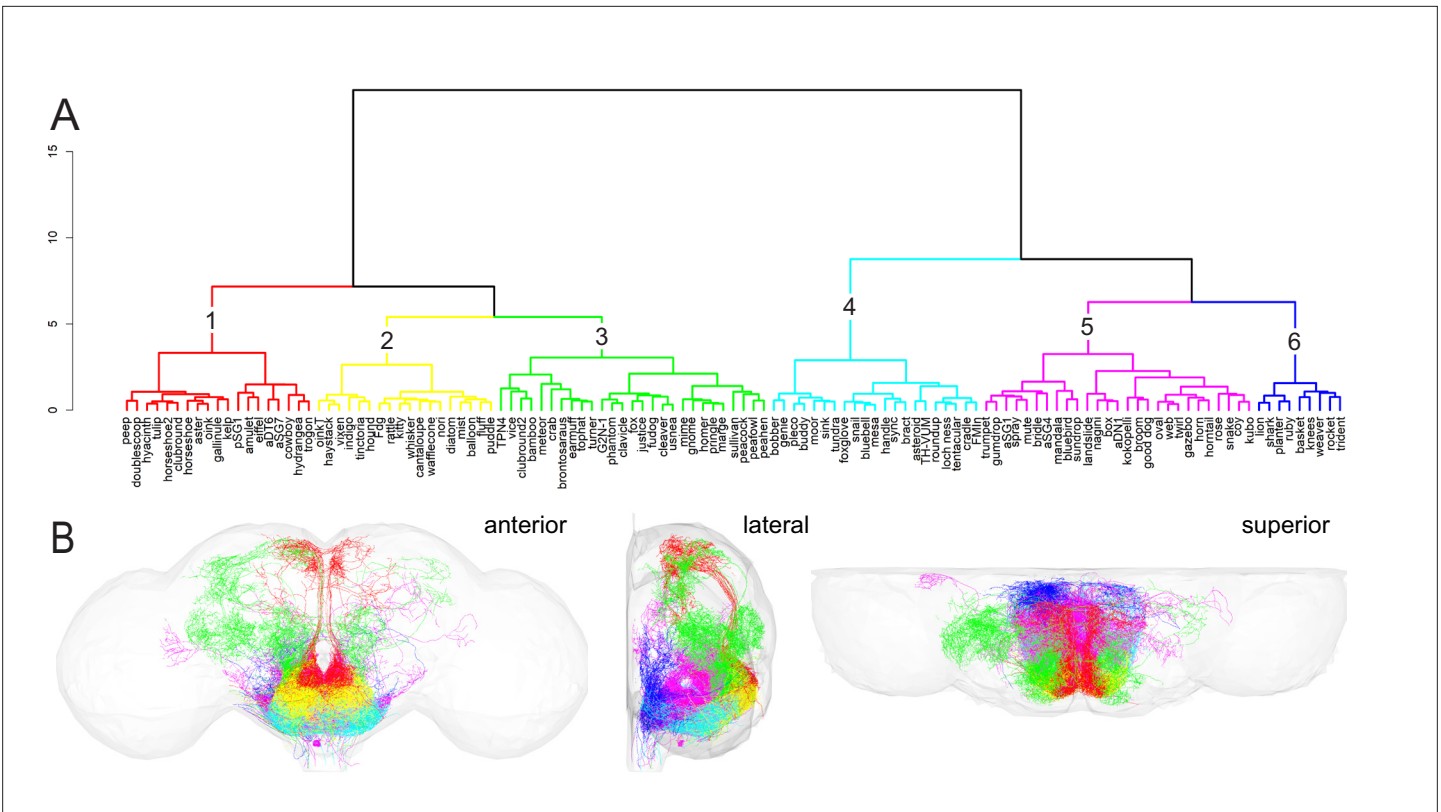

**Figure 3.** Hierarchical clustering of subesophageal zone (SEZ) neuronal cell types. (**A**) Clustering of SEZ neuron types with NBLAST reveals six distinct morphological groups: group 1: red; group 2: yellow; group 3: green; group 4: cyan; group 5: fuchsia; group 6: royal blue. Group number is indicated by the black number above each cluster. The vertical axis represents the distance or dissimilarity between the clusters. (**B**) Morphology of all neuron types in each cluster plotted according to the color code in (**A**). Central brain neuropil (gray) is plotted for reference. Anterior (left), lateral (middle), and superior (right) views are shown.

The online version of this article includes the following source data, source code, and figure supplement(s) for figure 3:

**Source code 1.** R code used for NBLAST clustering and visualization of cell type clusters.

**Source code 2.** Neuronlist object composed of dotprops, called by the NBLAST clustering code.

**Source code 3.** Template surface data file for visualizing brain neuropil, called in the NBLAST clustering code.

**Source data 1.** Metadata for the neuronlist in *Figure 3—source code 2*.

**Figure supplement 1.** Expression patterns of split-GAL4 lines targeting neuronal cell types not included in NBLAST clustering.

**Figure supplement 2.** Ward's joining cost and the differential of Ward's joining cost for hierarchical clustering of subesophageal zone (SEZ) neuronal cell types with NBLAST.

**Figure supplement 2—source data 1.** Source data for panel A.

**Figure supplement 2—source data 2.** Source data for panel B.

of the brain. In total, 121 of the 138 SEZ cell types targeted by the collection are represented in this dataset. The remaining cell types were excluded from NBLAST analysis because MCFO images were not available. The expression pattern of the best split-GAL4 line targeting each cell type excluded from NBLAST analysis is shown in *Figure 3—figure supplement 1*. After preprocessing, we computed an all-by-all similarity matrix for the represented cell types with NBLAST and hierarchically clustered the resulting NBLAST scores using Ward's method (*Costa et al., 2016*; *Figure 3B*). Ward's method is an agglomerative hierarchical clustering method that groups items into clusters that minimize within-cluster variance. Ward's joining cost, which is based on the variance of the data within a cluster, should increase significantly when distinct groups within the data are forced to join (*Braun et al., 2010*). Since the expected number of groups was not known beforehand, we analyzed Ward's joining cost and the differential of Ward's joining cost to quantitatively determine group number. We chose six groups due

to the low joining cost and the increase in the differential of Ward's joining cost when moving from six to five groups (*Figure 3—figure supplement 2*).

The resulting supergroups share anatomical similarities and coordinates that reveal that the SEZ is organized into layered and stacked domains. Five of the six supergroups are layered from anterior to posterior: 1 and 2 most anterior, followed by 3, 5, and finally 6 most posterior. Groups 1 and 2 are in a similar anterior plane but group 1 is positioned superior to group 2. Group 4 sits below these domains, wrapping the inferior surface of the SEZ. A lateral view illustrates that group 5 appears

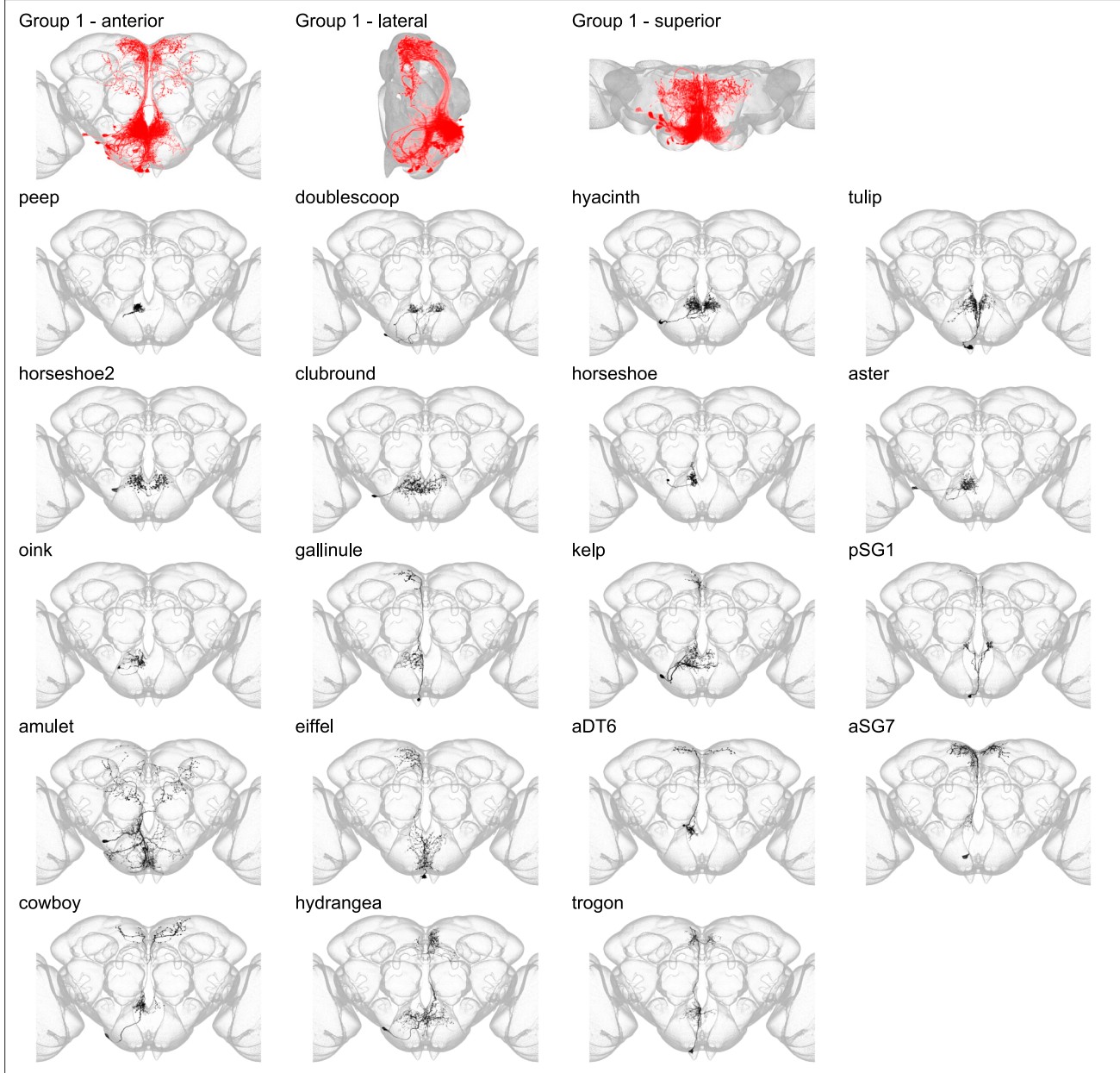

**Figure 4.** Morphology of neuron types in group 1. Segmented example images for each neuron type in group 1. The top row shows the morphology of all neuron types in group 1 (red) overlaid in the JRC 2018 unisex coordinate space (gray) in anterior, lateral, and superior views. Below, the morphology of individual group members is shown separately. Individual neuron morphology is shown in black while the outline of the JRC 2018 unisex template is shown in gray. In *Figures 3–8*, the segmented neurons were imaged with a 63× objective and registered to the full-size JRC 2018 unisex template. The optic lobes have been partially cropped out of each panel.

The online version of this article includes the following figure supplement(s) for figure 4:

**Figure supplement 1.** Expression patterns of the best split-GAL4 for each neuron type in roup 1.

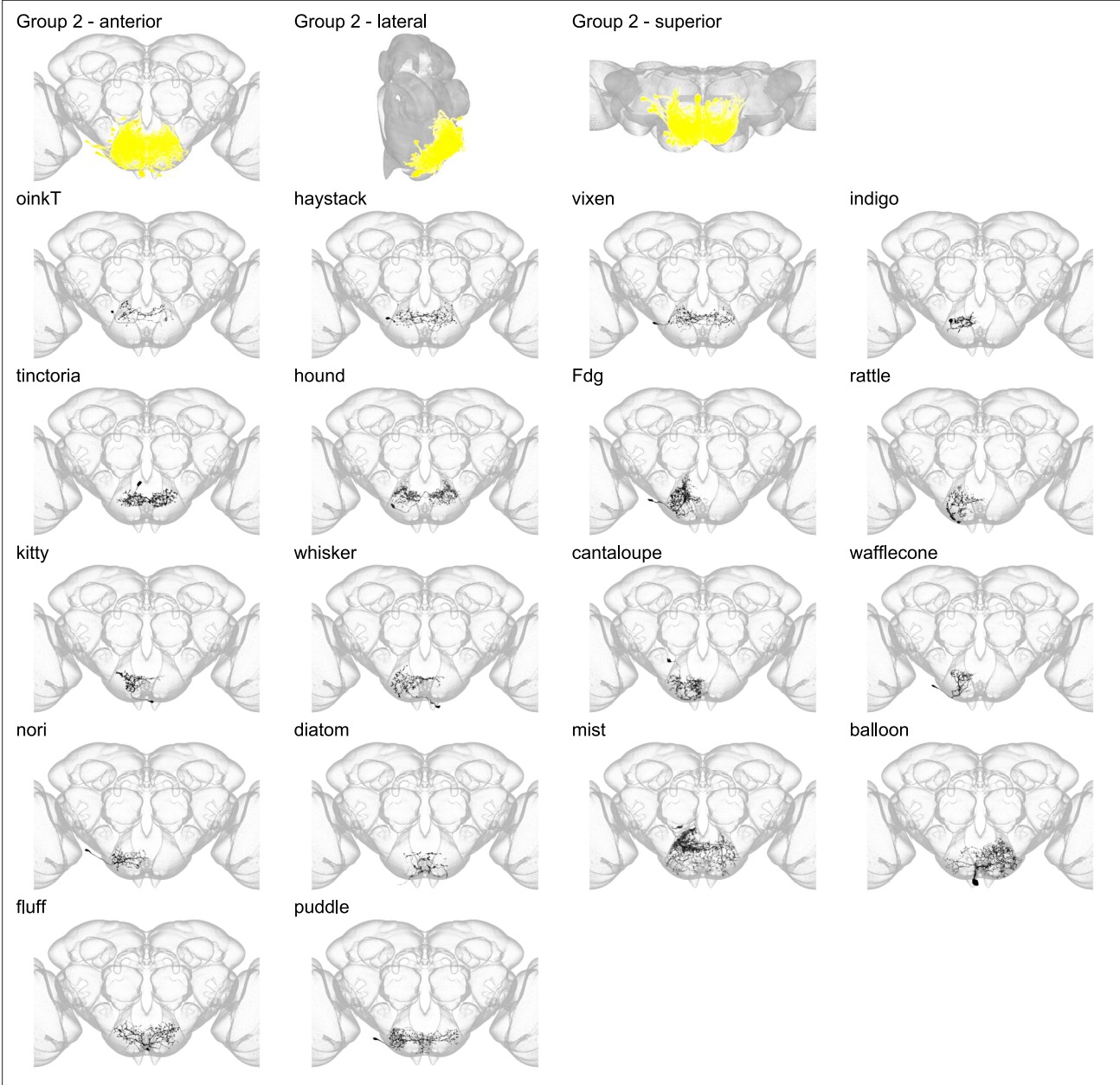

**Figure 5.** Morphology of individual neuron types in group 2. The top row shows the morphology of all neuron types in group 2 (yellow), with the morphology of individual group members shown below.

The online version of this article includes the following figure supplement(s) for figure 5:

**Figure supplement 1.** Expression patterns of the best split-GAL4 for each neuron type in group 2.

**Figure supplement 2.** Diatom morphology in the CNS and proboscis.

**Figure supplement 3.** Novel Fdg split-GAL4 lines label the previously identified Fdg cell type.

to form a 'roll' shape and is surrounded by group 3 anterior, group 4 inferior, and group 6 posterior. For each group, we show the morphology of an individual, segmented neuron for each cell type (*Figures 4–9*) as well as the pattern of the best split-GAL4 line for that cell type (*Figures 4–9*, *Figure 9—figure supplement 1*).

Group 1 is composed of neurons that arborize in the prow and flange (*Figure 4* and *Figure 4—figure supplement 1*), the superior, anterior, and medial regions of the SEZ. Based on their anatomical position, the 19 cell types that make up group 1 may originate from the tritocerebral neuromere.

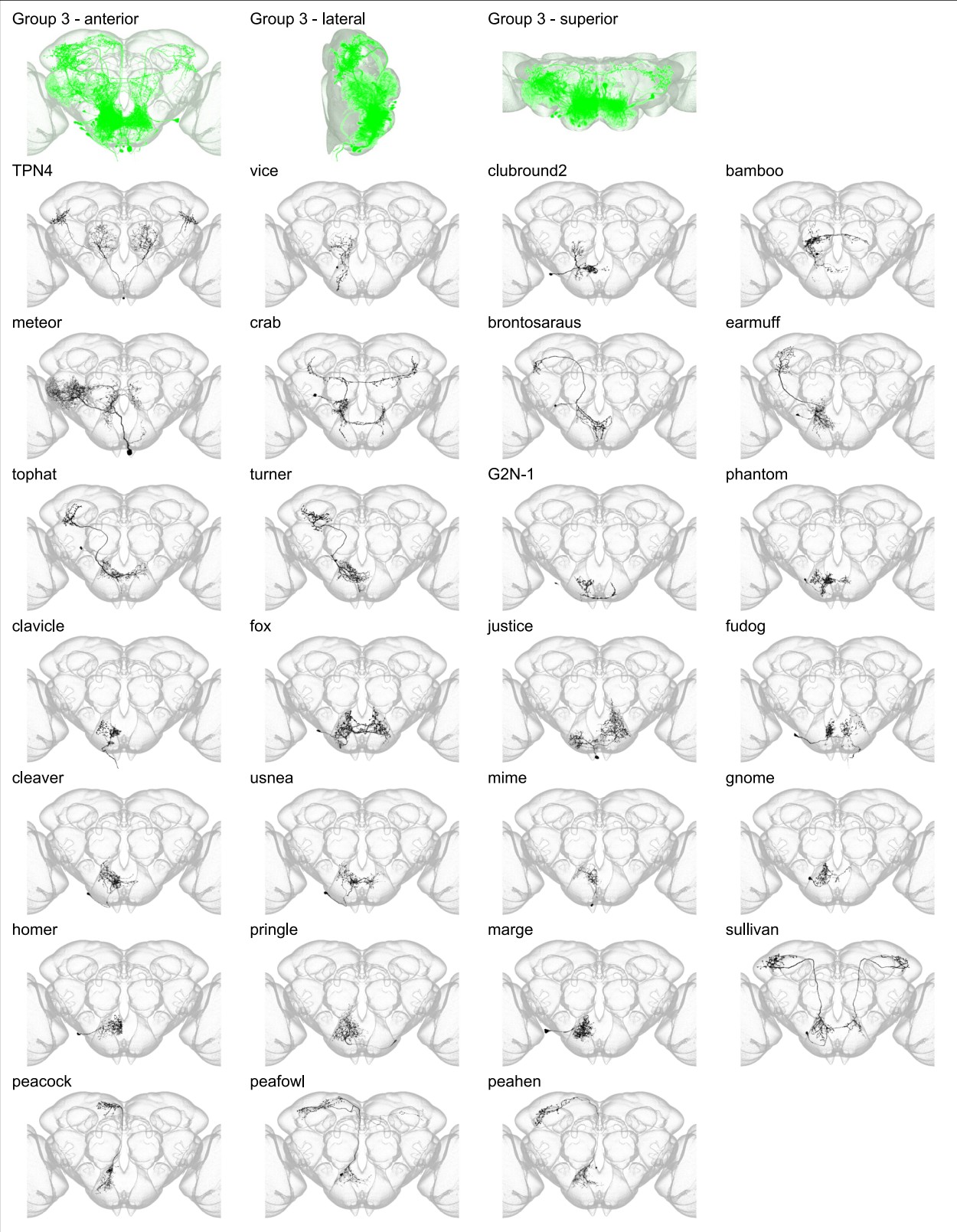

**Figure 6.** Morphology of individual neuron types in group 3. The top row shows the morphology of all neuron types in group 3 (green), with the morphology of individual group members shown below.

The online version of this article includes the following figure supplement(s) for figure 6:

**Figure supplement 1.** Expression patterns of the best split-GAL4 for each neuron type in group 3.

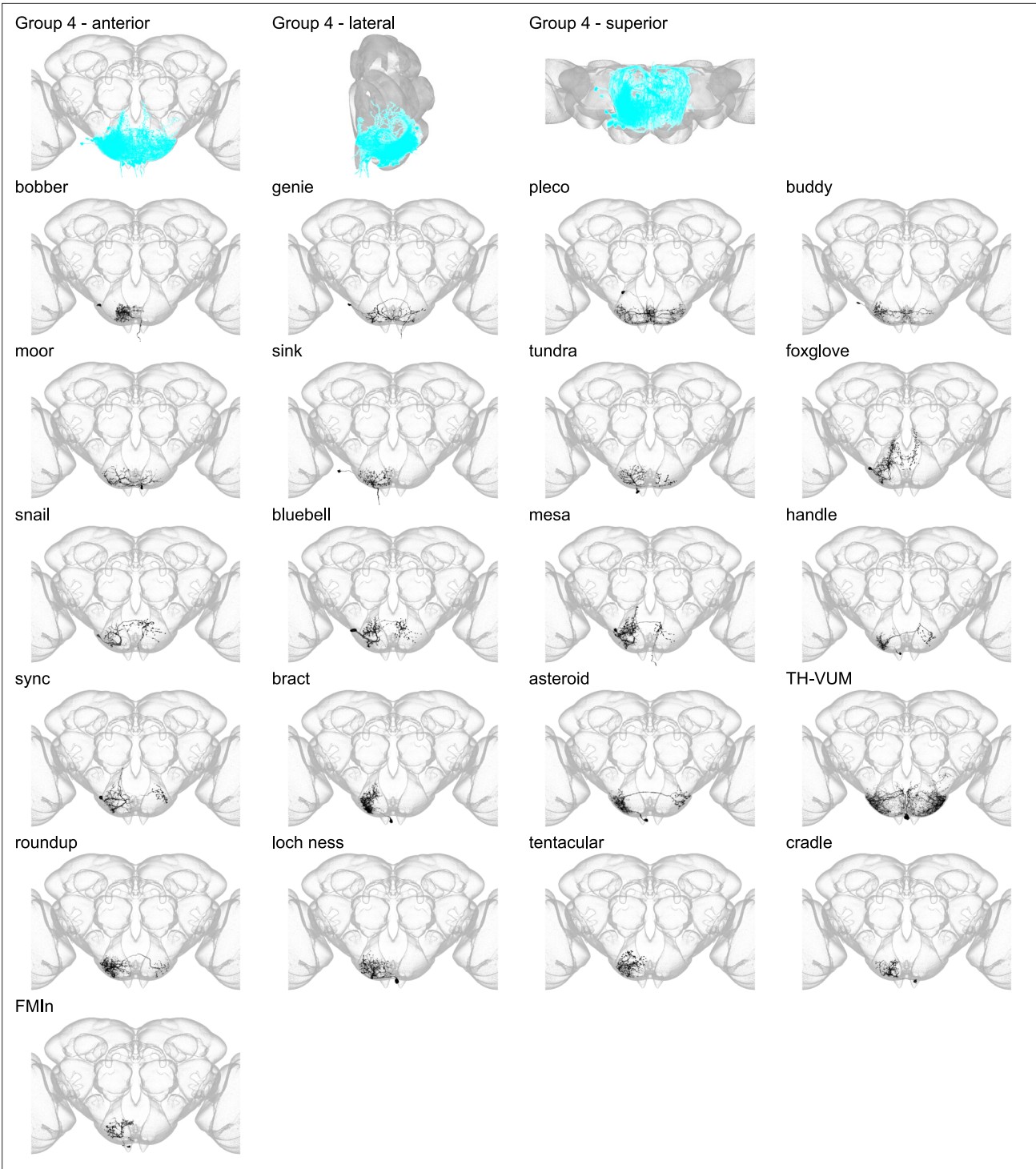

**Figure 7.** Morphology of individual neuron types in group 4. The first three panels show the morphology of all neuron types in group 4 (cyan), with the morphology of individual group members shown below.

The online version of this article includes the following figure supplement(s) for figure 7:

**Figure supplement 1.** Expression patterns of the best split-GAL4 for each neuron type in group 4.

**Figure supplement 2.** Axonal morphology of descending interneurons clustered into group 4.

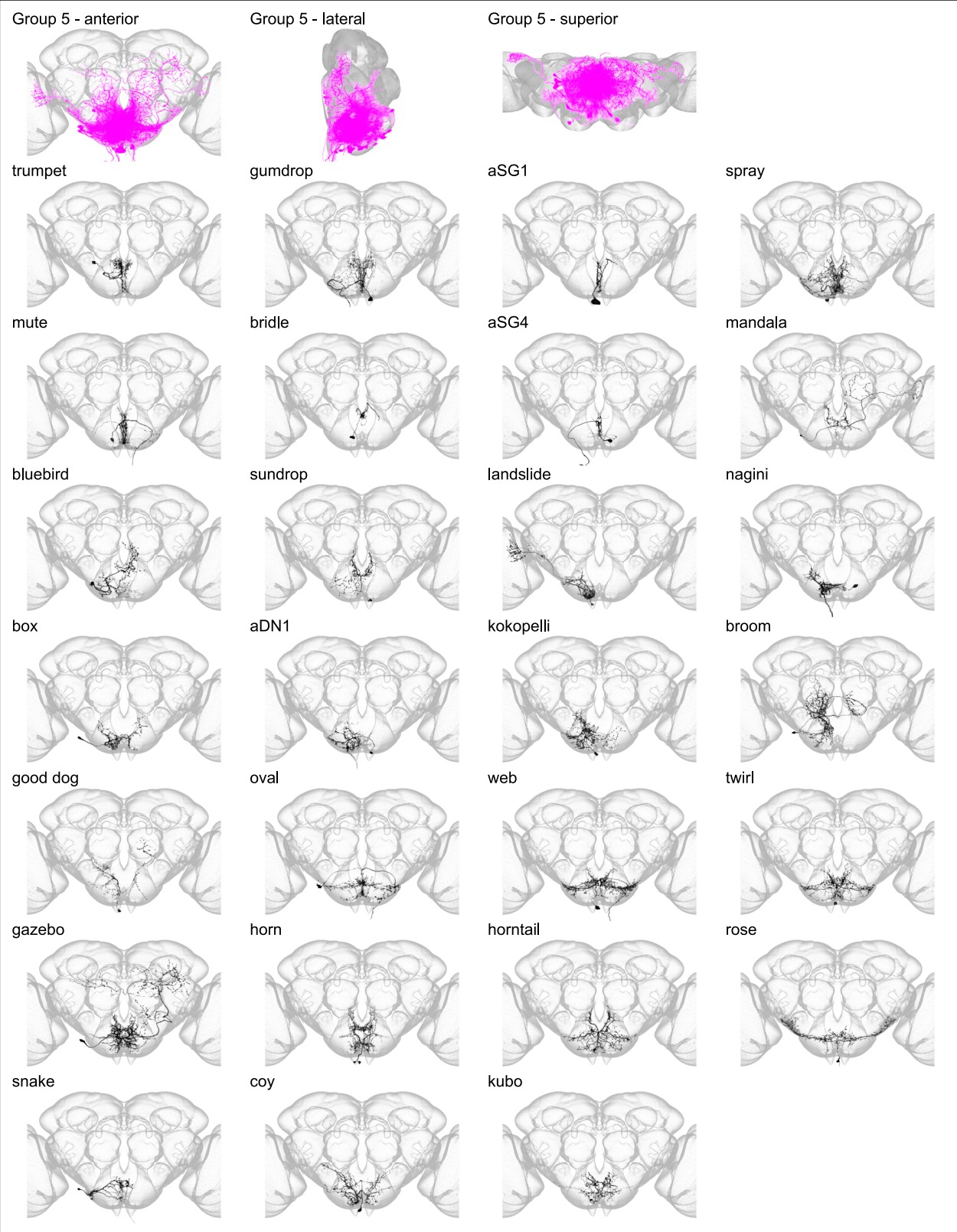

**Figure 8.** Morphology of individual neuron types in group 5. The top row shows the morphology of all neuron types in group 5 (fuchsia), with the morphology of individual group members shown below.

The online version of this article includes the following figure supplement(s) for figure 8:

*Figure 8 continued on next page*

Group 1 is neatly split into interneurons (peep, doublescoop, hyacinth, tulip, horseshoe2, clubround, horseshoe, aster, and oink) and projection neurons (gallinule, kelp, pSG1, amulet, eiffel, aDT6, aSG7, cowboy, hydrangea, and trogon). Notably, all projection neurons in group 1 send arbors to the superior medial protocerebrum (SMP). This group includes three previously morphologically described fruitless positive (Fru+) neuronal cell types: pSG1, aSG7, and aDT6 (*Liu, 2012*; *Jai et al., 2010*).

Group 2 contains only SEZ interneurons (*Figure 5* and *Figure 5—figure supplement 1*) plus one novel sensory neuron type with cell bodies in the proboscis labellum (diatom; *Figure 5—figure supplement 2*). Cell types in this group arborize in the anterior and superior region of the GNG, inferior to group 1. Of the 18 members of this group, only one cell type, 'feeding neuron' (Fdg), has been previously described (*Flood et al., 2013*). Fdg neurons respond to food presentation in starved flies and activation of Fdg induces a feeding sequence. The Fdg split-GAL4 lines reported here co-label the previously identified Fdg (*Figure 5—figure supplement 3*) and greatly improve specific genetic access.

Group 3 contains 27 members and is composed of SEZ projection neurons and interneurons that overlap with the dendrites of these projection neurons (*Figure 6* and *Figure 6—figure supplement 1*). In the SEZ, group 3 neurons arborize just anterior to the boundary between the anterior and posterior SEZ and in the superior region of the GNG, sometimes innervating the SAD or vest. Group 3 sits posterior to both groups 1 and 2 in the SEZ. In contrast to group 1 projection neurons, group

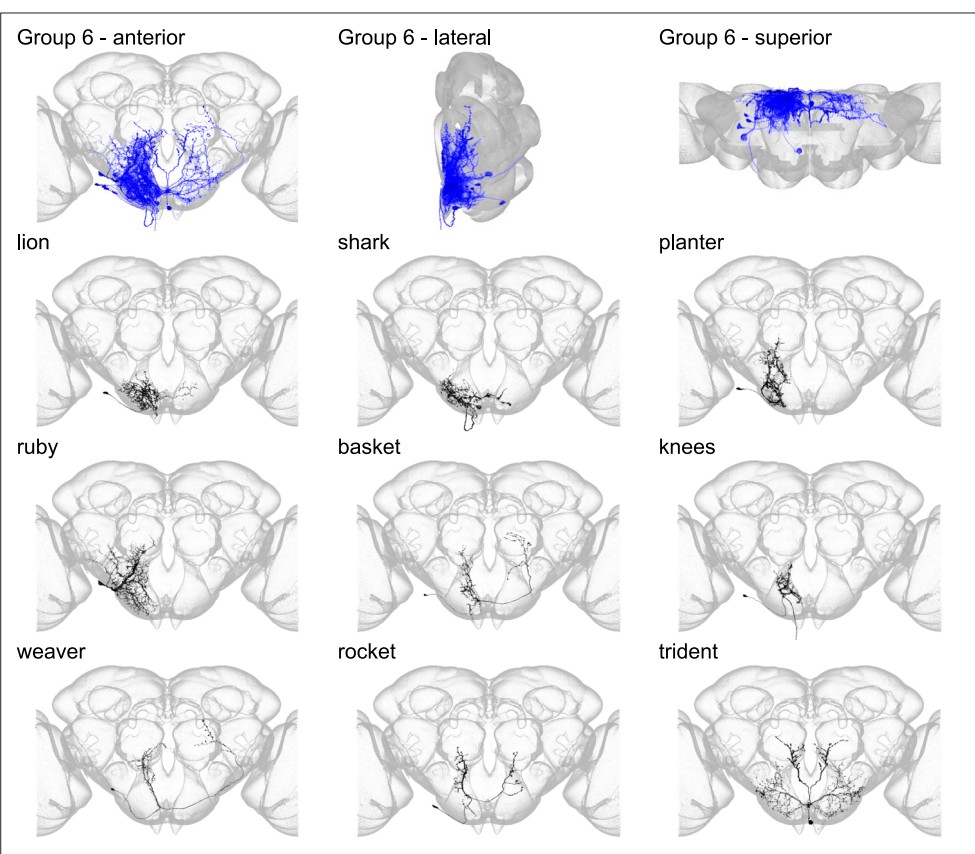

**Figure 9.** Morphology of individual neuron types in group 6. The top row shows the morphology of all neuron types in group 6 (royal blue), with the morphology of individual group members shown below.

The online version of this article includes the following figure supplement(s) for figure 9:

**Figure supplement 1.** Expression patterns of the best split-GAL4 for each neuron type in group 6.

2 projection neurons innervate diverse brain regions, including the superior lateral protocerebrum, superior clamp, and posteriorlateral protocerebrum, among others. One member of group 3 has been previously reported, gustatory second-order neuron type 1 (G2N-1) (*Miyazaki et al., 2015*).

Group 4 contains interneurons that arborize in the inferior GNG and wrap the inferior surface of the GNG (*Figure 7* and *Figure 7—figure supplement 1*). Group 4 sits inferior to groups 2 and 3. Notably, group 4 includes six novel DNs that arborize in the GNG and frequently descend to the leg neuropil (LegNp) in the prothoracic neuromere (*Figure 7—figure supplement 2*; *Court et al., 2020*). Of the 26 members of group 4, only one, tyrosine hydroxylase ventral unpaired medial (TH-VUM), has been previously reported. TH-VUM is a dopaminergic neuron that influences the probability of proboscis extension (*Marella et al., 2012*).

Group 5 contains 27 interneurons, projection neurons, and DNs that arborize just posterior to the boundary between the anterior and posterior SEZ, flanked by groups 3, 4, and 6 on the anterior, inferior, and posterior sides, respectively (*Figure 8* and *Figure 8—figure supplement 1*). Projection neurons in group 5 arborize in the lobula, inferior and superior clamp, and inferior bridge, among other regions. The seven DNs in this group send their axons most frequently to the leg neuropils and to the abdominal ganglion (Abd; *Figure 8—figure supplement 2*). Three members of group 5 have been previously reported. Two are previously described Fru+ neurons, aSG1 and aSG4 (*Jai et al., 2010*). The third neuronal type, aDN1, triggers antennal grooming when activated (*Hampel et al., 2015*).

Group 6 contains neurons in the posterior of the brain, spanning the GNG, inferior posterior slope, and superior posterior slope (*Figure 9* and *Figure 9—figure supplement 1*) and is the most posterior group in the SEZ. This small group contains only nine members, and group 6 cell types do not project to higher neuropils. One member, dubbed knees, is a DN that innervates neck neuropil and wing neuropil. No members of this morphological group have been previously reported.

## SEZ interneurons tend to have mixed polarity

To shed light on the structure of information flow both within the SEZ and out of the SEZ to the higher brain and VNC, we undertook polarity analysis of the 121 SEZ cell types that were segmented for NBLAST clustering analysis. These 121 cell types include 81 interneuron cell types, 26 projection neuron cell types, 13 DN cell types, and 1 sensory neuron cell type. We used both polarity staining with pre-synaptically localized HA-tagged Synaptotagmin and the smooth versus varicose appearance of neurites to score the presence of pre- and postsynaptic processes in each brain region in the central brain and VNC (*Court et al., 2020*; *Ito et al., 2014*; *Namiki et al., 2018*). Upon examination of many cell types, we found that SEZ cell types frequently lack a defined axon and dendrite. Instead, inputs and outputs are mixed on the same processes. We designated these cell types as possessing mixed polarity. Other cell types have mostly mixed polarity but still retain a distinct arbor region where synaptic outputs are concentrated. We termed this category of cell types to have biased polarity. A third category is polarized with clearly separated processes dedicated to either synaptic inputs or synaptic outputs. To supplement our annotation of the presence of axons, dendrites, or both in each neuropil compartment and polarization strategy, we also indicated whether each cell type is an interneuron, projection neuron, DN, or sensory neuron (*Figure 10*, left).

Among the SEZ interneuron cell types we analyzed, 39/81 (48%) have mixed polarity, 22/81 (27%) have biased polarity, and 20/81 (25%) are classically polarized. Interneuron types are distributed throughout the six cell type supergroups with most groups containing interneurons of all polarity classes. However, all group 2 interneurons (making up 17/18 cell types in group 2) have either biased or mixed polarity. This suggests that the interneurons of group 2 may participate in reciprocally connected circuits. Among interneuron cell types that are clearly polarized, there were some cases in which no axon was evident in the brain (including peep, shark, bridle, aSG1, and aSG4). In these cases, the presence of severed processes suggests that these cell types may not be interneurons and may instead send projections out of the central nervous system.

Most SEZ projection neurons analyzed in this study are clearly polarized (20/26, 77%). However, a few have biased (4/26, 15%) or mixed polarity (2/26, 8%). Polarized cell types belong to several cell type groups (3, 5, 6) and project to numerous brain regions including the lobula and the superior, inferior, and ventrolateral neuropils. In contrast, all projection neuron cell types with biased polarity have axons in the SMP and belong to group 1. The high proportion of clearly polarized SEZ projection

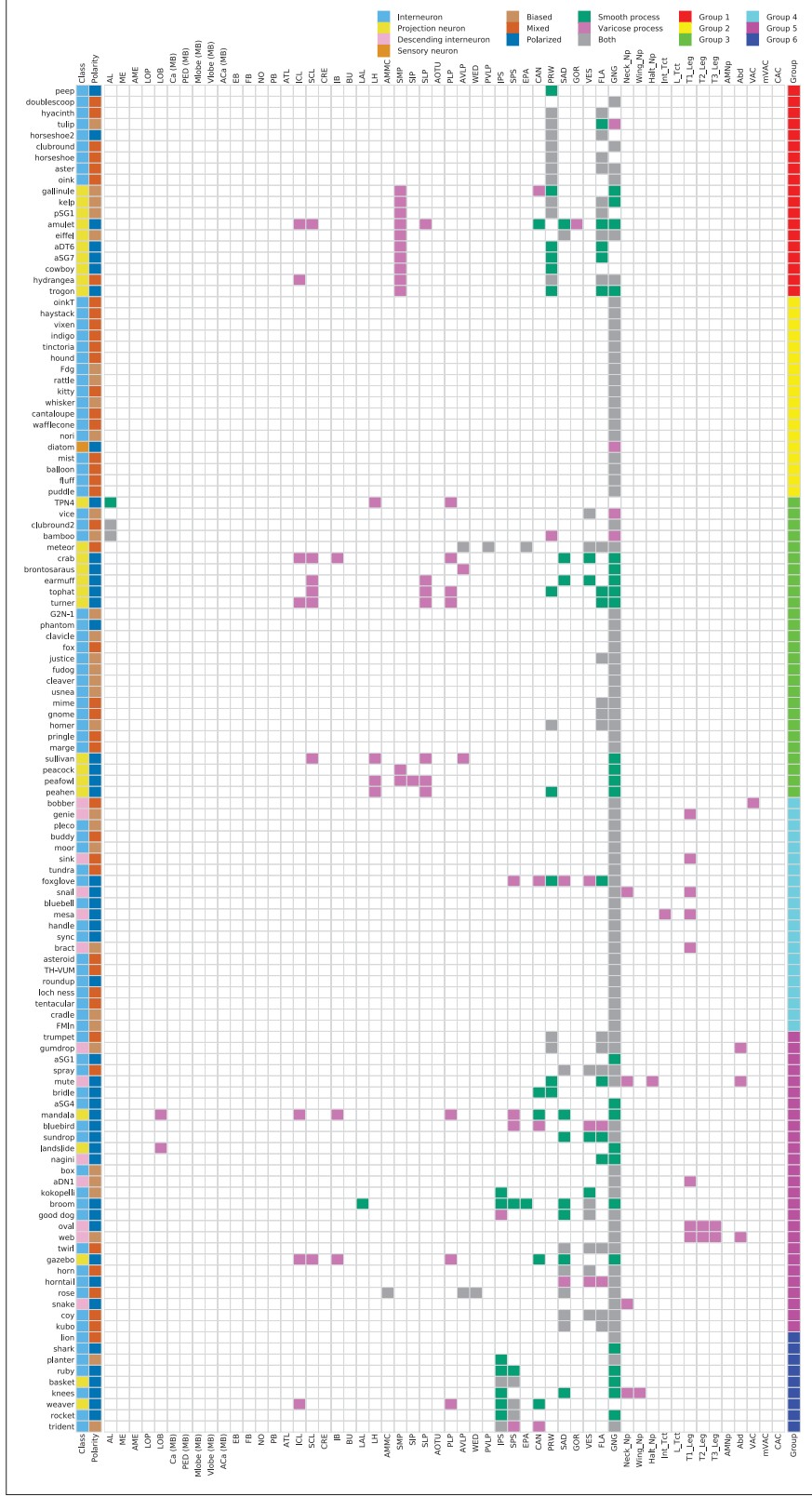

**Figure 10.** Innervation profile of subesophageal zone (SEZ) neuron types. (Leftmost column) Cell types are members of one of four cell type classes: interneuron (light blue), projection neuron (yellow), descending interneuron (light pink), or sensory neuron (orange). Interneurons are confined within the SEZ, while projection neurons project from the SEZ to higher neuropils in the central brain. Descending interneurons project their axons

*Figure 10 continued on next page*

*Figure 10 continued*

from the SEZ through the neck connective to the ventral nerve cord (VNC). Sensory neurons project their axons from elsewhere in the body into the SEZ. Class is indicated for each neuron type by the filled pixel to the right of each neuron type name. (Second-to-leftmost column) Neuron types are polarized in a biased (light brown), mixed (red-orange), or polarized (dark blue) manner. The polarity class for each neuron type is indicated. (Center) The innervation profile for each neuron type is indicated by the filled pixels in its corresponding row. Brain region abbreviations follow the definitions and naming conventions of *Ito et al., 2014* for the central brain and *Court et al., 2020* for the VNC. The locations of smooth processes (dendrites, green), varicose processes (axons, dark pink), or both smooth and varicose processes (axons and dendrites, gray) are indicated by defined neuropil region. VNC neuropil regions are grouped on the right of the figure. Innervation of the VNC was varicose in all cases. (Far right) Cell group as defined by NBLAST clustering is indicated for each cell type. Group 1: red; group 2: yellow; group 3: green; group 4: cyan; group 5: fuchsia; group 6: royal blue.

The online version of this article includes the following figure supplement(s) for figure 10:

**Figure supplement 1.** Example images of cell types in the mixed, biased, and polarized polarity classes.

neurons suggests that they commonly carry information unidirectionally from the SEZ to other brain regions. We did not identify any projection neurons that link the SEZ directly to the central complex or the mushroom body. In addition to projection neurons, the SEZ Split-GAL4 Collection includes one novel sensory neuron class (diatom) that is polarized, with dendrites in the proboscis labellum and SEZ axonal projections (*Figure 5* and *Figure 5—figure supplement 1*).

Analyzing the polarity of the novel SEZ DNs described here, we find that 5/13 (39%) have biased polarity, 2/13 (15%) have mixed polarity, and 6/13 (46%) are clearly polarized. These DNs have outputs in the neck, haltere, and leg neuropils (in all three neuromeres), and in the abdominal ganglion, consistent with the observation that GNG DNs are frequently connected with the leg neuropils (*Namiki et al., 2018*).

Thus, polarity analysis reveals that most types of SEZ interneurons have either mixed or biased polarity. In contrast, SEZ projection neurons are frequently clearly polarized while the SEZ DNs reported here are mostly mixed or polarized. These distinct polarization strategies may reflect the functional roles that interneurons, projection neurons, and DNs play in SEZ circuits. Interneurons in the SEZ may participate in reciprocally connected local networks while projection neurons and DNs may primarily relay information from the SEZ to other brain regions.

## Discussion

Here, we describe the SEZ Split-GAL4 Collection, a library of 277 split-GAL4 lines covering 138 SEZ cell types, which affords unprecedented genetic access to SEZ neurons for behavioral and functional study. Our studies provide insight into the diversity of SEZ cell types and their organization into discrete anatomical domains. The SEZ Split-GAL4 Collection will enable further investigation of how local SEZ circuitry and ascending SEZ paths process sensory inputs and control specific behaviors.

Most of the SEZ Split-GAL4 lines are specific, with 149/277 lines classified as ideal or excellent. These lines will be useful to manipulate individual SEZ cell types for behavioral, functional, and imaging experiments. The remaining, less specific, lines (those belonging to the good or poor categories) will still be useful for imaging and as starting points for creating more specific reagents. Good and poor lines may be used to generate CDM masks to search for new hemidrivers to make further split-GAL4 lines. Alternatively, their expression patterns may be refined using Killer Zipper or three-way intersections with LexA or QF lines (*Dolan et al., 2017*; *Shirangi et al., 2016*). All lines in the SEZ Split-GAL4 Collection may be used to generate further tools including complementary split-LexA and split-QF reagents (*Riabinina et al., 2019*; *Ting et al., 2011*). Split-LexA and split-QF lines may be used in concert with the split-GAL4 lines reported here to simultaneously manipulate two independent neuronal populations for advanced intersectional experiments, including behavioral epistasis.

By combining insights from a single-cell transcriptome atlas with direct cell counts of SEZ neuromeres, we estimate that the SEZ Split-GAL4 Collection labels 30% of the ~1700 neurons in the SEZ. Because of the lack of stereotyped neuronal cell body positions in *D. melanogaster*, it is not possible to assign cell bodies to defined neuropil regions without a genetic marker. The advantage of our method of estimating SEZ neuron number is that it is based on analysis of the four genetically

defined SEZ neuromeres, the tritocerebral, the mandibular, the maxillary, and the labial neuromeres. However, previous reports demonstrate that some deutocerebral commissures cross below the esophageal foramen, and therefore an unknown number of deutocerebral cell bodies may be part of the SEZ (*Boyan et al., 2003*; *Ito et al., 2014*). The limitations of our estimate of SEZ neuron number therefore include the inability to directly count cells derived from the tritocerebral neuromere, the inability to directly count neurons rather than glia, and the inability to assess deutocerebral contributions. Thus, our estimate of SEZ cell number is likely an underestimate. Once all SEZ neurons are densely reconstructed in an EM volume, direct counts of SEZ neuronal cell bodies obtained by EM will provide a more accurate assessment of SEZ neuron number. Regardless, the SEZ Split-GAL4 Collection targets 510 neuronal cell bodies, which represents a substantial improvement in our ability to precisely target SEZ cell types for functional and morphological analysis. We did not ascertain the neuromere or neuroblast of origin of the SEZ cell types in the SEZ Split-GAL4 Collection. However, recent work has established reliable anatomical criteria that define the boundaries between the four SEZ neuromeres and has mapped all secondary lineages of the SEZ (*Hartenstein et al., 2018*). Future efforts should focus on bridging previously identified fascicle, neuropil, and sensory domains into a common template or coordinate space to determine the neuromere and neuroblast origin of SEZ cell types.

Discovering and genetically targeting SEZ cell types required the use of registered light-level imagery and computer-assisted searching. We used four distinct strategies to identify 129 novel and 9 previously reported SEZ cell types in registered light-level imagery. Critically, each of these strategies allowed us to use CDM mask searching to identify additional hemidrivers with which to target each cell type of interest. CDM mask searching enabled combing of large datasets and greatly increased the ease and speed of split-GAL4 generation over previous methods (*Otsuna et al., 2018*). The same strategies can be leveraged to gain genetic access to yet-undiscovered SEZ cell types. The recent electron microscopy (EM) volumes of the *D. melanogaster* brain provide an avenue for identifying SEZ cell types that are not covered by the SEZ Split-GAL4 Collection. Notably, this approach awaits comprehensive reconstruction of the SEZ, a region that is not included in the recently published dense reconstruction of the 'hemibrain' volume (*Scheffer et al., 2020*). Another EM volume, 'FAFB,' provides imagery of an entire adult female fly brain at synaptic resolution and includes the SEZ (*Zheng et al., 2018*). Improvements in automated reconstruction of EM volumes coupled with large-scale human annotation should soon provide exhaustive reconstruction of the SEZ from which to identify additional SEZ cell types (*Dorkenwald et al., 2020*). Furthermore, available bridging registrations between EM volumes and light-level imagery should facilitate the identification of hemidrivers to target SEZ cell types discovered from EM reconstructions (*Bates et al., 2020*). Even without identifying additional SEZ cell types, the split-GAL4 reagents described will allow behavioral and functional evaluation of circuit hypotheses derived from EM imagery.

Our analyses of the SEZ Split-GAL4 Collection provide insight into the cellular architecture of the SEZ. To computationally probe the organization of the SEZ, we morphologically clustered 121 SEZ cell types using NBLAST (*Costa et al., 2016*). This approach reveals six cellular domains in the SEZ that are organized in a largely layered fashion from anterior to posterior. This layered structure is also hinted at by the recent description of SEZ neuropil domains throughout development from the larva to the adult (*Kendroud et al., 2018*). Based on anatomical position and the known function of a few SEZ neurons, it is tempting to speculate that different morphological clusters may participate in different behavioral functions. Group 1 contains projection neurons that innervate the region of the SMP surrounding the pars intercerebralis (PI), suggesting that group 1 neurons may impinge on neurosecretory neurons or function in energy and fluid homeostasis circuits. The proximity of group 1 interneurons to previously described interoceptive SEZ neurons (ISNs) (*Jourjine et al., 2016*) and ingestion neurons (IN1) (*Yapici et al., 2016*) supports this hypothesis. Group 2 contains Fdg, a feeding-related neuron, as well as cell types (indigo, tinctoria) that are located near pumping motor neurons (*Manzo et al., 2012*; *McKellar et al., 2020*), suggesting that group 2 neurons have roles in feeding sequence generation. Group 3 contains G2N-1, a candidate second-order gustatory neuron, and projection neurons that innervate recently described taste-responsive SLP regions (*Kim et al., 2017*; *Snell et al., 2020*), suggesting that group 3 may, in part, be composed of taste-responsive neurons. Many interneurons in group 4 are located near proboscis motor neurons that control rostrum protraction, haustellum extension, and labellar spreading (*Kendroud et al., 2018*; *McKellar et al., 2020*), indicating

that group 4 members function in proboscis motor control. The proximity of neurons in group 5 to previously described stopping neuron MAN (*Bidaye et al., 2014*), and the inclusion of an antennal grooming neuron, suggests that group 5 neurons may participate in circuits that control grooming and stopping behaviors. Group 6 is located in the posterior SEZ and posterior slope, regions implicated in flight behaviors, including wing and neck control (*Namiki et al., 2018*; *Robie et al., 2017*). While we hypothesize potential behavioral functions for each supergroup, we readily acknowledge that the roles of the neurons described here are likely more diverse.

Our studies also shed light on information flow both within the SEZ and out of the SEZ to the higher brain. We identified 91 local interneurons, 30 projection neurons, 16 descending neurons, and 1 sensory neuron. Polarity analysis of 121/138 of the SEZ cell types covered by the SEZ Split-GAL4 Collection revealed that SEZ interneurons tend to have mixed or biased polarity while SEZ projection neurons tend to be classically polarized. Polarity analyses of the lateral horn, mushroom body, descending neurons, and protocerebral bridge identified few neurons with completely mixed polarity (*Aso et al., 2014a*; *Aso et al., 2014b*; *Dolan et al., 2019*; *Namiki et al., 2018*; *Wolff and Rubin, 2018*). Unlike these brain regions, the SEZ contains a large number of local interneurons. The mixed polarity of the SEZ interneurons argues for local and reciprocal connectivity between neurons, with information flowing in networks rather than unidirectional streams. Projection neurons, in contrast, may serve chiefly to pass information from highly interconnected SEZ circuits to other brain regions in a unidirectional manner. Notably, we identified many SEZ projection neurons that innervate the SMP—a region known to contain neurosecretory cell types. This may betray a role for acute taste detection or feeding circuit activation in the regulation of hormone secretion. In addition, the frequent innervation of the superior lateral protocerebrum and lateral horn by SEZ projection neurons may hint at the site of olfactory-gustatory synthesis. In contrast, we did not identify projection neurons that link the SEZ directly to the central complex or mushroom body. If dense reconstruction of EM volumes corroborates the lack of direct connectivity between the SEZ and these regions, information must be conveyed through indirect pathways. As an example, taste information influences local search behaviors during foraging, a task that is expected to involve the central complex (*Haberkern et al., 2019*). Indirect relay of taste information to the central complex to inform foraging behavior would be consistent with previous anatomical studies suggesting that the central complex receives diverse indirect sensory inputs (*Pfeiffer and Homberg, 2014*). Furthermore, the mushroom body is known to respond to taste, raising the possibility that taste information from gustatory sensory neuron axons in the SEZ must be relayed through yet another brain region before reaching mushroom body cell types (*Harris et al., 2015*). Thus, our analysis of SEZ neuron polarity indicates local SEZ processing and demonstrates direct pathways to a subset of higher brain regions.

Overall, the SEZ Split-GAL4 Collection represents a valuable resource that will facilitate the study of the SEZ. Our analysis of the collection reveals the cellular anatomy and polarity of individual SEZ neurons and their organization into six discrete domains in the SEZ. Coupled with emerging insights from reconstruction of EM volumes, the SEZ Split-GAL4 Collection will allow the use of genetic dissection to test circuit-level hypotheses about sensory processing and motor control in the SEZ.

## Materials and methods

Key resources table

| Reagent type (species) or resource | Designation | Source or reference | Identifiers | Additional information |
|---|---|---|---|---|
| Genetic reagent (*Drosophila melanogaster*) | Polarity reporter, w; +; 3xUAS-Syt:: smGFP-HA (su(Hw)attP1), 5xUAS-IVS- myr::smGFP-FLAG (VK5) | *Aso et al., 2014b* | | |
| Genetic reagent (*D. melanogaster*) | csChrimson Reporter/Optogenetic effector, 20xUAS- csChrimson::mVenus in attP18 | *Klapoetke et al., 2014* | BDSC:55134; FLYB:FBst0055134 | |
| Genetic reagent *D. melanogaster* | UAS-Syt-HA;; | *Robinson et al., 2002* | | Recombined with 20XUAS-CsChrimson-mVenus trafficked in attP18 when used for polarity analysis experiments |

*Continued on next page*

*Continued*

| Reagent type (species) or resource | Designation | Source or reference | Identifiers | Additional information |
|---|---|---|---|---|
| Genetic reagent (*D. melanogaster*) | pBPhsFLP2:PEST in attP3; 13xLexAop2-> dSTOP>-myr::smGFP-OLLAS in su(Hw)attP5, 13xLexAop2-> dSTOP>-myr::smGFP-V5 in attP40/CyO; 13xLexAop2-> dSTOP>-myr::smGFP-FLAG in attP2/TM2 | This work | | LexA-based MCFO line with heat shock flippase |
| Genetic reagent (*D. melanogaster*) | R57C10-Flp2::PEST in su(Hw)attP8;; pJFRC201-10XUAS-FRT>STOP>FRT-myr::smGFP-HA in VK00005,pJFRC240-10XUAS-FRT>STOP >FRT-myr::smGFP-V5-THS-10XUAS-FRT>STOP>FRT-myr::smGFP-FLAG in su(Hw)attP1/TM2 | *Nern et al., 2015* | BDSC:64089; FLYB:FBst0064089 | Short name: MCFO-3 |
| Genetic reagent (*D. melanogaster*) | pBPhsFlp2::PEST in attP3;; pJFRC210-10XUAS-FRT>STOP>FRT-myr::smGFP-OLLAS in attP2, pJFRC201-10XUAS-FRT>STOP>FRT-myr::smGFP-HA in VK0005, pJFRC240-10XUAS-FRT>STOP >FRT-myr::smGFP-V5-THS-10XUAS-FRT>STOP>FRT-myr::smGFP-FLAG in su(Hw)attP1/ TM2 | *Nern et al., 2015* | BDSC:64086; FLYB:FBst0064086 | Short name: MCFO-2 |
| Genetic reagent (*D. melanogaster*) | ;;Dfd-LexA | *Simpson, 2016* | | |
| Genetic reagent (*D. melanogaster*) | ;;Scr-LexA | *Simpson, 2016* | | |
| Genetic reagent (*D. melanogaster*) | Labial-GAL4 | *Hirth et al., 2001* | BDSC:43652; FLYB:FBst0043652 | |
| Genetic reagent (*D. melanogaster*) | ;LexAop-nls-GCaMP6s in VIE-260b; | This work | | |
| Genetic reagent (*D. melanogaster*) | ;;His2Av-mRFP | *Pandey et al., 2005* | BDSC:23650; FLYB:FBst0023650 | |
| Genetic reagent (*D. melanogaster*) | ; UAS-Syn21-nlsGCaMP6s-p10 in VIE-260b; | This work | | |
| Genetic reagent (*D. melanogaster*) | ;;UAS-His::mRFP | *Emery et al., 2005* | FLYB:FBtp0022240 | |
| Genetic reagent (*D. melanogaster*) | ;81E10-LexAp65 in JK22C; | This work | | Approach and promoter have been previously described (*Jenett et al., 2012*; *Pfeiffer and Homberg, 2014*) |
| Genetic reagent (*D. melanogaster*) | NP883-GAL4 | *Yoshihara and Ito, 2000* | Kyoto:103803; FLYB:FBst0302671 | Line in which Fdg was originally identified (*Flood et al., 2013*) |
| Genetic reagent (*D. melanogaster*) | NP5137-GAL4 | *Yoshihara and Ito, 2000* | Kyoto:113602; FLYB:FBst0316329 | Line which also labels Fdg (*Flood et al., 2013*) |
| Genetic reagent (*D. melanogaster*) | 13XLexAop2-CsChrimson-tdT (attP18), 20XUAS-IVS-Syn21-opGCaMP6f p10 (Su(Hw)attp8);; | *Morimoto et al., 2020* | | |
| Antibody | Anti-Brp (mouse monoclonal) | DSHB, University of Iowa, USA | DSHB Cat# nc82, RRID:AB_2314866 | (1:40) |
| Antibody | Anti-GFP (chicken polyclonal) | Thermo Fisher Scientific | Thermo Fisher Scientific Cat# A10262, RRID:AB_2534023 | (1:1000) |
| Antibody | Anti-dsRed (rabbit polyclonal) | Takara | Takara Bio Cat# 632496, RRID:AB_10013483 | (1:1000) |
| Antibody | Anti-chicken Alexa Fluor 488 (goat polyclonal) | Thermo Fisher Scientific | Thermo Fisher Scientific Cat# A-11039, RRID:AB_2534096 | (1:1000) |
| Antibody | Anti-rabbit Alexa Fluor 568 (goat polyclonal) | Thermo Fisher Scientific | Thermo Fisher Scientific Cat# A-11036, RRID:AB_10563566 | (1:1000) |
| Antibody | Anti-mouse Alexa Fluor 647 (goat polyclonal) | Thermo Fisher Scientific | Thermo Fisher Scientific Cat# A-21236, RRID:AB_2535805 | (1:500) |

*Continued on next page*

*Continued*

| Reagent type (species) or resource | Designation | Source or reference | Identifiers | Additional information |
|---|---|---|---|---|
| Software, algorithm | VVDviewer | *Otsuna et al., 2018* | RRID:SCR_021708 | https://github.com/JaneliaSciComp/VVDViewer |
| Software, algorithm | Fiji | *Schindelin et al., 2012* | RRID:SCR_002285 | http://fiji.sc/ |
| Software, algorithm | Computational Morphometry Toolkit | *Rohlfing and Maurer, 2003* | RRID:SCR_002234 | https://www.nitrc.org/projects/cmtk/ |
| Software, algorithm | R Project for Statistical Computing | *R Development Core Team, 2018* | RRID:SCR_001905 | https://www.r-project.org/ |
| Software, algorithm | NeuroAnatomy Toolbox | *Jefferis and Manton, 2014* | 10.5281/zenodo.1136106, RRID:SCR_017248 | http://jefferis.github.io/nat/ |
| Software, algorithm | Ilastik | *Berg et al., 2019* | RRID:SCR_015246 | https://www.ilastik.org/ |
| Software, algorithm | MaMuT Plugin | *Wolff et al., 2018* | | https://imagej.net/MaMuT |
| Software, algorithm | Janelia WorkStation | *Rokicki et al., 2019* | RRID:SCR_014302 | https://doi.org/10.25378/janelia.8182256.v1 |

## *Drosophila* husbandry

All experiments and screening were carried out with adult *D. melanogaster* females raised at 25 °C on standard *Drosophila* food. Adult females were mated and dissected within 1 week of eclosion. Construction of stable split-GAL4 lines was performed as previously described (*Dionne et al., 2018*).

## Anatomical directional terms and neuropil nomenclature

Throughout this resource, we refer to anatomical directional terms according to the body axis as previously defined (*Court et al., 2020*; *Ito et al., 2014*). The central brain and SEZ are shown in all figures as seen from the anterior side of the brain with the superior up, unless otherwise indicated. *Figures 3–9* show views of the central brain from the anterior with the superior side up (labeled 'anterior'), views from the (fly's) right lateral side with the superior side up (labeled 'lateral'), and views from the superior side with the ventral side up (labeled 'superior') to show the three-dimensional morphology of the six supergroups. The VNC is always displayed from the inferior view with the anterior side up. Directional terms for the proboscis are also indicated according to the body axis in *Figure 5—figure supplement 2*. We also refer to neuropil regions and their corresponding abbreviations according to established and published nomenclature (*Court et al., 2020*; *Ito et al., 2014*).

## Counting SEZ neurons

Either Dfd-LexA or Scr-LexA was crossed to a reporter line with LexAop-nls-GCaMP6s (this work) and His2Av-mRFP (*Pandey et al., 2005*). Labial-GAL4 was crossed to a reporter line with UAS-nls-GCaMP6s (this work) and UAS-His2Av-mRFP (*Emery et al., 2005*). Brains dissected as described (https://www.janelia.org/project-team/flylight/protocols, 'Dissection and Fixation 1.2% PFA').

The following primary antibodies were used:

- 1:40 mouse α-Brp (nc82) (DSHB, University of Iowa, USA)
- 1:1000 chicken α-GFP (Invitrogen A10262)
- 1:1000 rabbit α-dsRed (Takara, Living Colors 632496)

The following secondary antibodies were used:

- 1:500 α-mouse AF647 (Invitrogen, A21236)
- 1:1000 α-chicken AF488 (Life Technologies, A11039)
- 1:1000 α-rabbit AF568 (Invitrogen, A21236)

Immunohistochemistry was carried out as described (https://www.janelia.org/project-team/flylight/protocols, 'IHC-Anti-GFP') substituting the above antibodies and eschewing the pre-embedding fixation steps. Ethanol dehydration and DPX mounting was carried out as described (https://www.janelia.org/project-team/flylight/protocols, 'DPX Mounting'). Images were acquired with a Zeiss LSM 880 NLO AxioExaminer at the Berkeley Molecular Imaging Center. A Plan-Apochromat 63× /1.4 Oil DIC

M27 objective was used at zoom 1.0. Acquired images had a voxel size of 0.132 μm × 0.132 μm × 0.500 μm. Expression in the SEZ was imaged in a tiled fashion and then stitched in Fiji using the 'Grid/Collection stitching' plugin with 'Unknown Positions' and 'Linear Blending' (*Preibisch et al., 2009*). Example overview images shown in *Figure 1* and *Figure 1—figure supplement 1* were also acquired with a Zeiss LSM 880 NLO AxioExaminer at the Berkeley Molecular Imaging Center. A Plan-Apochromat 25× /0.8 Imm Corr DIC M27 objective was used at zoom 0.7. Acquired images had a voxel size of 0.474 μm × 0.474 μm × 0.886 μm. Example overview images were acquired for visualization only and were not used for cell counting as described below.

SEZ cell number was quantified with Ilastik using the 'Pixel Classification' and 'Object Classification' workflows (*Berg et al., 2019*). The pixel classifier was trained to segment only cell bodies expressing both LexAop-nls-GCaMP6s and His2Av-mRFP, which improved pixel and object classification accuracy when compared to using LexAop-nls-GCaMP6s without His2Av-mRFP (data not shown). Then, to verify counts derived from automated Ilastik quantification, manual ground truth counts of example image regions (four subregions each for Dfd-LexA and Scr-LexA) were compared to counts of the same regions derived from Ilastik. Ground truth counts were carried out in three dimensions with the MaMuT plugin in Fiji (*Wolff et al., 2018*). Error was calculated at 0.5% for Dfd-LexA images and –1.4% for Scr-LexA images.

## Split-GAL4 intersections

Novel SEZ cell types were identified using the following strategies:

1. Visual search through several large, publicly available GAL4 collections designed to tile the nervous system (*Jenett et al., 2012*; *Pfeiffer et al., 2008*; *Tirian and Dickson, 2017*).
2. LexA-based MCFO of Scr-LexA and Dfd-LexA. In total, 232 Scr-LexA samples and 320 Dfd-LexA samples were examined.
3. MCFO (*Nern et al., 2015*) screening of subsets of the Janelia Research Campus and Vienna Tile GAL4 collections that have dense SEZ expression in which individual cell morphologies were difficult to parse (*Meissner et al., 2020*). 66,080 CDM images from MCFO of 2182 unique lines were examined.
4. Re-registration of open-access images of individual SEZ cell types from MARCM screens of broad GAL4 drivers, available on FlyCircuit (*Chiang et al., 2011*). 22,598 female samples re-registered to the 'JFRC 2010' template (*Jenett et al., 2012*) were analyzed.

Following identification of cell types, we created representative CDM masks and used CDM mask searching (*Otsuna et al., 2018*) to find additional enhancers whose expression patterns seemed to include the desired cell type. We annotated all drivers that putatively drove expression in each of the identified cell types. We searched the following CDM images: 27,534 CDM images covering 6575 Janelia Research Campus GAL4 lines; 18,047 CDM images covering 8031 GAL4 Vienna Tile lines; and 66,080 CDM images from MCFO of 2182 unique Janelia Research Campus and Vienna Tile lines. In total, we used 86,861 CDM images for CDM mask searching. We then assessed the availability of hemidrivers for each of the enhancers (ADs and DBDs). The split-GAL4 hemidrivers used in this study were previously generated at Janelia Research Campus (*Dionne et al., 2018*; *Tirian and Dickson, 2017*). Then, the expression patterns for all possible AD-DBD combinations for a given cell type were screened. Screening was carried out in adult female flies as previously described (*Dionne et al., 2018*). A single female central nervous system was screened per combination. With few exceptions, screening was carried out by FlyLight using the FLyLight split-screen protocol: (https://www.janelia.org/project-team/flylight/protocols, 'IHC-Adult Split Screen'). Following dissection, staining, and mounting, split-GAL4 combinations were screened by eye using epifluorescence on an a LSM710 confocal microscope (Zeiss) with a Plan-Apochromat 20× /0.8 M27 objective. We assessed the specificity of each line in the central nervous system only, not in peripheral tissues. Imagery was viewed and organized using the Janelia Workstation (*Rokicki et al., 2019*). Useful combinations with limited SEZ expression were selected for initial confocal imaging using a 20× objective. Following imaging, useful combinations were further sorted and annotated in a custom database. The resulting database of SEZ split-GAL4 lines contains the following:

- Target cell type
- AD and DBD
- Unique SS identifier

- Line quality (ideal > excellent > good> poor)
- A text description of any off-target expression
- Types of imagery collected, including polarity and MCFO data

After stabilization (*Dionne et al., 2018*), select split-GAL4 lines were further characterized. We selected at least one split-GAL4 line per cell type for detailed documentation, including polarity staining (to assess expression pattern in multiple central nervous systems and to determine the location of synaptic outputs), MCFO characterization, and 63× imaging. Polarity staining was carried out by crossing stabilized split-GAL4 lines to either w; +; 3xUAS-Syt::smGFP-HA(su(Hw)attP1), 5xUAS-IVS-myr::smGFP-FLAG (VK5) or UAS-Syt-HA, 20XUAS-CsChrimson-mVenus (attP18);;. When crossed to w; +; 3xUAS-Syt::smGFP-HA(su(Hw)attP1), 5xUAS-IVS-myr::smGFP-FLAG (VK5) dissection and staining were carried out by FlyLight according to the FlyLight 'IHC-Polarity Sequential' protocol (https://www.janelia.org/project-team/flylight/protocols). When crossed to 20XUAS-CsChrimson-mVenus (attP18);; dissection and staining were carried out by FlyLight according to the FlyLight 'IHC-Polarity Sequential Case 5' protocol (https://www.janelia.org/project-team/flylight/protocols). MCFO characterization of stable split-GAL4 lines was accomplished by crossing stable lines to MCFO-2 or MCFO-3 (see Key resources table for full genotypes). If crossed to MCFO-2, adult flies were heat shocked at 37 ° C for either 30 or 60 min 1 day after eclosion. Dissection and staining of MCFO samples were carried out by FlyLight according to the FlyLight MCFO staining protocol (https://www.janelia.org/project-team/flylight/protocols, 'IHC-MCFO'). Samples stained for polarity and MCFO analysis were first imaged on an a LSM710 confocal microscope (Zeiss) with a Plan-Apochromat 20× /0.8 M27 objective. Then, sample images were viewed using the Janelia Workstation (*Rokicki et al., 2019*) and several samples per line were chosen for higher-resolution imaging. Higher-resolution imaging of select samples was carried out on a LSM710 confocal microscope (Zeiss) with a Plan-Apochromat 63× /1.40 oil immersion objective. If multiple tiles were required to cover the region of interest, tiles were stitched together (*Yu and Peng, 2011*).

## Co-labeling experiments with Fdg lines

To label with Fdg with a binary expression system that is independent of GAL4/UAS, we created a LexA line from the 81E10 promoter region (*Jenett et al., 2012*; *Pfeiffer and Homberg, 2014*) inserted into the JK22C attP site (*Knapp et al., 2015*). The JK22C site was chosen to mitigate the possibility of transvection between transgenes inserted in identical attP sites on homologous chromosomes (*Mellert and Truman, 2012*). We created a stock carrying 13XLexAop2-CsChrimson-tdtomato in attP18 and 20XUAS-IVS-Syn21-opGCaMP6f p10 in su(Hw)attP8 recombined on the X chromosome (*Morimoto et al., 2020*) and 81E10-LexA in JK22C. To perform co-labeling experiments, we crossed this stock to either NP883, NP5137 (*Flood et al., 2013*; *Yoshihara and Ito, 2000*), SS31345, SS46913, or SS46914. Dissection, staining, and mounting were carried out as described in the 'Counting SEZ neurons' section above. Images were acquired with a Zeiss LSM 880 NLO AxioExaminer at the Berkeley Molecular Imaging Center. A Plan-Apochromat 63× /1.4 Oil DIC M27 objective was used at zoom 0.7. Acquired images had a voxel size of 0.188 μm × 0.188 μm × 1.000 μm.

## Morphological clustering with NBLAST

63× MCFO images were registered to the full-size JRC 2018 unisex template (*Bogovic et al., 2020*) using CMTK (https://www.nitrc.org/projects/cmtk). A single example of each cell type targeted by the collection was selected for segmentation in VVDviewer (https://github.com/takashi310/VVD_Viewer; *Otsuna et al., 2018*). The following 17 cell types covered by the SEZ Split-GAL4 Collection were excluded because suitable MCFO images were not available: bay, bower, braces, bubbA, bump, clownfish, handup, linea, mothership, oinkU, pampa, portal, seagull, slink, spirit, stand, and willow. The expression pattern of the best split-GAL4 line for each excluded cell type is shown in *Figure 3—figure supplement 1*. The remaining 121 cell types covered by the collection were included in NBLAST analysis. Registration quality was assessed by viewing the overlap between the template brain and the registered nc82 reference channel to ensure that selected images were well registered. Further, selected images were only used if the morphology of the cell type of interest was clearly visible and not intermingled with other cells or neuronal processes that might lead to false merges or truncations due to neighboring cell types. Images were manually segmented in VVDviewer to remove non-specific background and other, clearly distinct cells. Following segmentation, images

were thresholded using the 'Huang' method (*Huang and Wang, 1995*), flipped to the right hemisphere of the brain, and scaled to a final voxel size of (x) 0.3766 × (y) 0.3766 × (z) 0.3794. Scaled images were then skeletonized with the 'Skeletonize 2d/3d' Fiji plugin (*Lee et al., 1994*). Skeletonized, scaled images were hierarchically clustered using NBLAST and Ward's method (*Costa et al., 2016*). This was carried out with the natverse toolkit in R (*Bates et al., 2020*). Group number was chosen by assessing Ward's joining cost and the differential of Ward's joining cost after *Braun et al., 2010*. Images of the resulting morphological clusters were further visualized in R, again using natverse (*Figure 3*). Catalog figures were assembled using full-sized segmented imagery in VVDviewer (*Figures 3–8*).

### Polarity analysis

Full-size registered, segmented example neuron images (prior to scaling or skeletonizing) created as described above were compared against established neuropil regions (*Court et al., 2020*; *Ito et al., 2014*) in VVDviewer. The presence of smooth versus varicose processes was scored after *Namiki et al., 2018*. Images from polarity staining were referenced where available.

### Acknowledgements

We thank the Janelia Fly Facility (Todd Laverty, Amanda Cavallaro, Scarlett Harrison, Karen Hibbard, Jui-Chun Kao, and Guillermo Gonzalez, among others) for help with fly husbandry and fly line generation. The FlyLight Project Team (https://www.janelia.org/project-team/flylight, Geoffrey Meissner, Kelley Lee, Zachary Dorman, Oz Malkesman, and others) performed brain dissections, immunohistochemistry, and confocal imaging for split-GAL4 screening and characterization after stabilization. We would like to acknowledge Geoffrey Meissner for testing and providing the recombined UAS-Syt-HA, 20XUAS-CsChrimson-mVenus trafficked in attP18 stock for split-GAL4 polarity analysis. Confocal imaging for the SEZ cell counting experiments was conducted at the CRL Molecular Imaging Center, supported by the Helen Wills Neuroscience Institute and NSF DBI-1041078. We would also like to thank Holly Aaron and Feather Ives for their microscopy training and assistance. Léo Guignard generously provided feedback and guidance without which automated cell counting with Ilastik would not have been possible. Ryo Minegishi shared code and tips for implementing NBLAST on new datasets. We would also like to thank Masayoshi Ito for sharing split-GAL4 combinations that labeled SEZ neurons.

## Additional information

### Funding

| Funder | Grant reference number | Author |
|---|---|---|
| Howard Hughes Medical Institute | | Gabriella R Sterne<br>Hideo Otsuna<br>Barry J Dickson |
| National Institute of Diabetes and Digestive and Kidney Diseases | F32DK117671 | Gabriella R Sterne |
| National Institute of General Medical Sciences | R01NS110060 | Gabriella R Sterne<br>Kristin Scott |

The funders had no role in study design, data collection and interpretation, or the decision to submit the work for publication.

### Author contributions

Gabriella R Sterne, Conceptualization, Data curation, Formal analysis, Funding acquisition, Investigation, Software, Visualization, Writing - original draft; Hideo Otsuna, Formal analysis, Methodology, Resources, Software, Writing - review and editing; Barry J Dickson, Kristin Scott, Conceptualization, Funding acquisition, Project administration, Supervision, Writing - review and editing

Author ORCIDs
Gabriella R Sterne ⓘ http://orcid.org/0000-0002-7221-648X
Barry J Dickson ⓘ http://orcid.org/0000-0003-0715-892X
Kristin Scott ⓘ http://orcid.org/0000-0003-3150-7210

Decision letter and Author response
Decision letter https://doi.org/10.7554/eLife.71679.sa1
Author response https://doi.org/10.7554/eLife.71679.sa2

---

## Additional files

### Supplementary files
• Supplementary file 1. Subesophageal zone (SEZ) Split-GAL4 Collection database. Each row of the database describes an individual split-GAL4 line generated in this study. The targeted cell type is noted in the 'Cell type' column. The 'AD' and 'DBD' columns note the AD and DBD hemidrivers that compose each line. The unique stable split code identifies each line and can be found in the 'SS number' column. 'SEZ Split-GAL4 Collection code' provides a shorter, unique identifier for each split-GAL4 line that is specific to this study. 'Quality: (P)oor, (G)ood, (E)xcellent, or (I)deal' column ranks the quality of each line, while 'Quality notes' provides a short, qualitative description of off-target expression in each line. The 'Imagery available: (P)olarity (M)CFO or (B)oth' notes which types of image data are available for each line at https://splitgal4.janelia.org/.

• Transparent reporting form

### Data availability
Detailed information about the split-GAL4s and available imagery is included in a supplemental database (Supplementary file 1). Image data are publicly available and all lines may be ordered at http://splitgal4.janelia.org.

The following previously published datasets were used:

| Author(s) | Year | Dataset title | Dataset URL | Database and Identifier |
|---|---|---|---|---|
| Davie K, Janssens J, Koldere D, De Waegeneer M, Pech U, Kreft , Aibar S, Makhzami S, Christiaens V | 2018 | A single-cell transcriptome atlas of the ageing *Drosophila* brain | https://scope.aertslab.org/#/Davie_et_al_Cell_2018/Davie_et_al_Cell_2018%2FAerts_Fly_AdultBrain_Filtered_57k.loom/gene | SCope, Davie_et_al_Cell_2018%2FAerts_Fly_AdultBrain_Filtered_57k.loom |

---

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
