## [Decision Letter]

**Acceptance summary:**

We believe that your work represents an important resource of information and tools to the *Drosophila* community. Specifically, we appreciate the careful anatomical characterisation of the SEZ area, one of the less well characterised brain regions in the central brain of insects. We anticipate that your work will be of great interest and help to researchers interested in understanding the representation of taste as well as the control of movement.

**Decision letter after peer review:**

Thank you for submitting your article "Classification and genetic targeting of cell types in the primary taste and premotor center of the *Drosophila* brain" for consideration by *eLife*. Your article has been reviewed by 3 peer reviewers, one of whom is a member of our Board of Reviewing Editors, and the evaluation has been overseen by Ronald Calabrese as the Senior Editor. The following individual involved in review of your submission has agreed to reveal their identity: Kristen Lee (Reviewer #2).

Essential Revisions (for the authors):

1) Please address the recommendations of the reviewers to improve figure clarity and presentation.

2) The reviewers suggest additional citations and additional discussion points. Please address these by adding appropriate citations and paragraphs in the text.

3) You will see that some additional experiments were suggested. Please consider adding additional experiments, if possible.

*Reviewer #1 (Recommendations for the authors):*

– Line 78: delete "of"

– Line 92-98: I appreciate the authors being candid about the limitations here.

– Line 111: why were cell types from AMMC not included? Perhaps can be explicit here.

– Line 152: maybe I missed this, but I was wondering what the source of the SEZ neuron number estimates is.

– Lines 276-279: I think this paragraph can be omitted in the Results section, since the same points are made in the Discussion and can be somewhat repetitive.

– Line 295: to me it seems more logical to have the quotation marks around "classically" rather than "polarized" to follow along with "mixed" and "biased" polarity. Later the authors use "clearly" polarized, so maybe say clearly polarized already in this line for consistency.

– Lines 336-339: This sentence can be revised for additional clarity.

– Line 564 show -> "shown".

– Figure 1: I think changing one set of color schemes will be useful, instead of using the same set of colors to label regions and quality of the gal4 line.

– Figure 2: is the vertical axis and tick marks meaningful in A? Also, perhaps enlarging the group #'s can help with visibility.

– Figure 6 supplement 2 is misplaced.

– Figure 4 supp 2: Please include the split line in the figure.

*Reviewer #2 (Recommendations for the authors):*

Result comments:

Ito, 2014 refers to all structures below the level of the esophagus as SEZ. They also annotate the cantle (CAN) and flange (FLA) as a periesophageal neuropil. Why are they not listed an anatomical regions of the SEZ on line 75? Especially since the flange is mentioned on line 207.

The Results section would greatly improve by mentioning in some more detail how the different cell types (i.e. names listed in Figure 1G, 2, 4-8) were determined, since they are a main focus of the figures, but rarely mentioned in the text. I found this discrepancy to be a little confusing/distracting.

Although the polarity data is interesting, I am curious as to whether this is a surprising finding or not? I am not acutely familiar with that field, and think some more speculation on this in the discussion or results would enhance reader understanding. What are the impacts of this result?

Figure comments:

Figure 1D, E: the labeling is confusing. I think the figure would benefit from describing what each color in the confocal image represents. There also seems to be a lot of noise from connective tissue/the neck connective which is a little distracting. If irrelevant, could this be stitched out?

Figure 1 – Supplement 1: Could the SEZ or esophagus be marked with a dash line? This will better orient the reader with the anatomical regions. Could this figure also be flipped to match the orientation of Figure 1 A, D, E?

Figure 1G feels out of place in Figure 1, and may fit better in Figure 2.

Figure 9 could benefit from a supplemental figure showing examples of each polarity type.

*Reviewer #3 (Recommendations for the authors):*

A) Suggestions for improved or additional experiments:

In analysis like this, there´s always the issue of where one should draw the line in terms of providing some/any kind of physiological/functional data, whether these are stainings with functionally relevant markers (e.g., neurotransmitters), behavioral studies, functional imaging, etc. In this case, there appears to be some instances where it´s screaming out: e.g., the IN1 or FDG. Both of these neurons have been published with extensive functional data, and it would be great to see if these are the same neurons indeed and to confirm it with some functional analysis.

B) Recommendation for improving the writing and presentation (in no particular order):

1) The authors mention different degree of line cleanness, and say rightly that some are more appropriate for behavioral analysis, others sufficient for imaging. However, these are based on stainings from CNS. Is is known how many of these "CNS clean" lines presented here label peripheral tissues, whether neuronal or non-neuronal tissues?

2) Nomenclature: two types of names are given here, one is the JRC number and the other is a "private" name. How would the nomenclature proceed in subsequent analysis, when for example, a second neuron which looks very similar to one presented here is identified? Would this be considered as belonging to the same class (such as seven neurons of the insulin producing cell cluster; or between these and the neighboring neurons of the PI). There is one named FDG-medial, invoking FDG; on what basis is this given this, rather than a completely separate name). This is not a criticism of the current naming at all, just wondering how it will play out as the SEZ coverage increases in the future, and more closely related/looking neurons become identified.

3) A major section is given to analysis of neuronal polarity based on bioinformatic analysis. Some selected illustrative examples would help (e.g., taking the "known" ones, like FDG, IN1, TH-VUM, or special sensory neuron they put more emphasis on, like "diatom"), and show in real anatomy what terms like mixed, biased and polarized actually look like.

4) This paper is focused on adult SEZ, and appropriately refers to earlier works in adults; but in some instances, it refers to analysis from embryo/larva (Kendroud et al., 2018; Hartenstein et al., 2018); subdivisions of the SEZ based on synaptic connectivity have been described in larva (Miroschnikow et al., 2018) but not mentioned here. Based on this, perhaps the authors can comment on what parts of the larval data are or are not relevant for their own work, or clearly state the works of adult vs larva or change the wording a bit (e.g., lines 45-46 in its current form concerning the cellular anatomy part).

---

## [Author Response]

Reviewer #1 (Recommendations for the authors):– Line 78: delete "of"

done

– Line 92-98: I appreciate the authors being candid about the limitations here.

Thank you

– Line 111: why were cell types from AMMC not included? Perhaps can be explicit here.

The AMMC is an anatomically segregated region. Other researchers are generating split-Gal4s for this region. We noted and justified not including the AMMC on lines 150-152.

– Line 152: maybe I missed this, but I was wondering what the source of the SEZ neuron number estimates is.

These estimates come from the first section of the results.

– Lines 276-279: I think this paragraph can be omitted in the Results section, since the same points are made in the Discussion and can be somewhat repetitive.

We deleted this paragraph.

– Line 295: to me it seems more logical to have the quotation marks around "classically" rather than "polarized" to follow along with "mixed" and "biased" polarity. Later the authors use "clearly" polarized, so maybe say clearly polarized already in this line for consistency.

We agree that the quotation marks are inconsistent. We removed the quotation marks for increased clarity.

– Lines 336-339: This sentence can be revised for additional clarity.

We edited this sentence.

– Line 564 show -> "shown".

corrected

– Figure 1: I think changing one set of color schemes will be useful, instead of using the same set of colors to label regions and quality of the gal4 line.

Based on all reviewers’ comments, we removed 1G from Figure 1 and included a new Figure 2 to describe the quality of the split-GAL4 lines.

– Figure 2: is the vertical axis and tick marks meaningful in A? Also, perhaps enlarging the group #'s can help with visibility.

The vertical axis represents the distance or dissimilarity between the clusters, noted in the figure legend. We have enlarged the group numbers to make them more visible.

– Figure 6 supplement 2 is misplaced.

corrected

– Figure 4 supp 2: Please include the split line in the figure.

Figure 5 —figure supplement 2 now includes the expression pattern of the split-GAL4 line.

Reviewer #2 (Recommendations for the authors):Result comments:Ito, 2014 refers to all structures below the level of the esophagus as SEZ. They also annotate the cantle (CAN) and flange (FLA) as a periesophageal neuropil. Why are they not listed an anatomical regions of the SEZ on line 75? Especially since the flange is mentioned on line 207.

The CAN and FLA are not considered to be part of the SEZ since they are lateral to and not inferior to the esophageal foramen, as stated in pp. 51 of the Supplemental Information of Ito, 2014: “In certain insect species, some of the neuropils in the PENP lie below the esophagus and have therefore been regarded as parts of the SEZ (or, in previous publications, the SEG). In *Drosophila*, the saddle, AMMC, and prow lie below the esophagus and can therefore be considered as parts of the SEZ (but not of the GNG).”

The Results section would greatly improve by mentioning in some more detail how the different cell types (i.e. names listed in Figure 1G, 2, 4-8) were determined, since they are a main focus of the figures, but rarely mentioned in the text. I found this discrepancy to be a little confusing/distracting.

We have included a description of how cell types were determined in the Results section (lines 138-142 and lines 158-163)

Although the polarity data is interesting, I am curious as to whether this is a surprising finding or not? I am not acutely familiar with that field, and think some more speculation on this in the discussion or results would enhance reader understanding. What are the impacts of this result?

We have added the following discussion on polarity (lines 446-458): “Our studies also shed light on information flow both within the SEZ and out of the SEZ to the higher brain. We identified 82 local interneurons, 26 projection neurons, 12 descending neurons, and 1 sensory neuron. Polarity analysis of the 121 SEZ cell types revealed that SEZ interneurons tend to have mixed or biased polarity while SEZ projection neurons tend to be classically polarized. Polarity analyses of the lateral horn, mushroom body, descending neurons, and protocerebral bridge identified few neurons with mixed polarity (refs). Unlike these brain regions, the SEZ contains a large number of local interneurons. The mixed polarity of the SEZ interneurons argues for local and reciprocal connectivity between neurons, with information flowing in networks rather than unidirectional streams. Projection neurons, in contrast, may serve chiefly to pass information from highly interconnected SEZ circuits to other brain regions in a unidirectional manner.”

Figure comments:Figure 1D, E: the labeling is confusing. I think the figure would benefit from describing what each color in the confocal image represents. There also seems to be a lot of noise from connective tissue/the neck connective which is a little distracting. If irrelevant, could this be stitched out?

We replaced the stitched Dfd and Scr example images with lower resolution example overview images of the samples used to count cell number (Figure 1D and E). We describe the colors in the legend but due to the reuse of colors in the figure we think that adding a color legend in the figure itself may be confusing.

Figure 1 – Supplement 1: Could the SEZ or esophagus be marked with a dash line? This will better orient the reader with the anatomical regions. Could this figure also be flipped to match the orientation of Figure 1 A, D, E?

We now include asterisks to mark the location of the esophageal foramen in Figure 1D and E as well as in Figure 1 —figure supplement 1. Furthermore, we have rotated the image in Figure 1 —figure supplement 1 to match the orientation of Figure 1 D and E.

Figure 1G feels out of place in Figure 1, and may fit better in Figure 2.

To address this, we moved Figure 1G to a new Figure 2 that is dedicated to describing the quality of the lines in the collection.

Figure 9 could benefit from a supplemental figure showing examples of each polarity type.

Thank you for your suggestion. We have included a new figure, Figure 10 —figure supplement 1 showing two examples of cell types belonging to each polarity class.

Reviewer #3 (Recommendations for the authors):A) Suggestions for improved or additional experiments:In analysis like this, there´s always the issue of where one should draw the line in terms of providing some/any kind of physiological/functional data, whether these are stainings with functionally relevant markers (e.g., neurotransmitters), behavioral studies, functional imaging, etc. In this case, there appears to be some instances where it´s screaming out: e.g., the IN1 or FDG. Both of these neurons have been published with extensive functional data, and it would be great to see if these are the same neurons indeed and to confirm it with some functional analysis.

We appreciate this suggestion and include data showing that three of the new Fdg split-GAL4 lines overlap with Fdg in the original lines by co-labeling studies (new Figure 5 —figure supplement 3). Although we confirmed the identity of Fdg, functional studies are beyond the scope of the study. We did not make a split-GAL4 for IN1.

B) Recommendation for improving the writing and presentation (in no particular order):1) The authors mention different degree of line cleanness, and say rightly that some are more appropriate for behavioral analysis, others sufficient for imaging. However, these are based on stainings from CNS. Is is known how many of these "CNS clean" lines presented here label peripheral tissues, whether neuronal or non-neuronal tissues?

We did not assess the expression of the lines outside of the central nervous system with the exception a line for the novel cell type diatom (SS40945), for which we looked at expression in the proboscis. We have clarified this in the methods (lines 593-594).

2) Nomenclature: two types of names are given here, one is the JRC number and the other is a "private" name. How would the nomenclature proceed in subsequent analysis, when for example, a second neuron which looks very similar to one presented here is identified? Would this be considered as belonging to the same class (such as seven neurons of the insulin producing cell cluster; or between these and the neighboring neurons of the PI). There is one named FDG-medial, invoking FDG; on what basis is this given this, rather than a completely separate name). This is not a criticism of the current naming at all, just wondering how it will play out as the SEZ coverage increases in the future, and more closely related/looking neurons become identified.

We envision that NBLAST searches and co-labeling studies will be essential to determine if neurons are the same or different in different lines. The question of whether neurons belong to the same class is interesting but will not be resolved by anatomical studies alone. As the tools available for EM cell type analysis progress, it is possible that our naming scheme will be supplanted by standardized cell type names based on comprehensive reconstruction of all cells in the SEZ. Regarding the similarity of cell type names to previously reported neurons, we agree with the reviewer that the names FDG-medial and IN2 are not appropriate and have renamed them to neutral names (FDG-medial to bract and IN2 to aster) that do not suggest similarities to previously studied neurons.

3) A major section is given to analysis of neuronal polarity based on bioinformatic analysis. Some selected illustrative examples would help (e.g., taking the "known" ones, like FDG, IN1, TH-VUM, or special sensory neuron they put more emphasis on, like "diatom"), and show in real anatomy what terms like mixed, biased and polarized actually look like.

We have now included a new figure (Figure 10 —figure supplement 1) showing examples of different neural polarities, including Fdg, G2N-1, TH-VUM, and diatom. We also show examples of cell types in the mixed, biased, and polarized classes. Our apologies again for the confusion between IN1 and IN2. We did not make any split-GAL4 lines for IN1 but named IN2 based on its similar location in the brain. We have renamed IN2 “aster” to avoid this confusion for future readers.

4) This paper is focused on adult SEZ, and appropriately refers to earlier works in adults; but in some instances, it refers to analysis from embryo/larva (Kendroud et al., 2018; Hartenstein et al., 2018); subdivisions of the SEZ based on synaptic connectivity have been described in larva (Miroschnikow et al., 2018) but not mentioned here. Based on this, perhaps the authors can comment on what parts of the larval data are or are not relevant for their own work, or clearly state the works of adult vs larva or change the wording a bit (e.g., lines 45-46 in its current form concerning the cellular anatomy part).

We appreciate this suggestion, added the omitted reference, and altered the wording (lines 49-50).